


# Particle and front tracking in experimental and computational avalanche dynamics

Michael Neuhauser[1], Anselm Köhler[1], Anna Wirbel[1], Felix Oesterle[1], Wolfgang Fellin[2], Johannes Gerstmayr[3], Falko Dressler[4], and Jan-Thomas Fischer[1]

[1]Department for Natural Hazards, Austrian Research Centre for Forests
[2]Department of Geotechnics, University of Innsbruck
[3]Department of Mechatronics, University of Innsbruck
[4]School of Electrical Engineering and Computer Science, Technical University Berlin

**Correspondence:** Jan-Thomas Fischer (jt.fischer@bfw.gv.at)

**Abstract.** Understanding particle motion in snow avalanches is essential for unravelling the driving processes behind transport phenomena and mobility. Our approach to investigating avalanche dynamics at the particle level combines data from a novel inflow sensor system, the AvaNodes, with radar measurements and simulation results from the thickness integrated flow module of AvaFrame, the open avalanche framework. The radar measurements offer a comprehensive view of the avalanche, serving as a reference for the AvaNodes' trajectories within it. This synthesis provides a holistic overview of the motion of avalanche particles and the front. The utilized com1DFA module in AvaFrame, equipped with a numerical particle grid method, enables a direct implementation of numerical particle tracking functionalities, facilitating a comparison between measurements and simulations. This unique combination prompts questions about the comparability of simulations and measurements on a particle level, yielding new insights into the thickness integrated model's ability to replicate real-scale snow avalanche particle behaviour assuming a modified Voellmy friction relation. Our work also highlights current limitations of comparing radar measurements and synthetic particle sensor systems with numerical simulation particles. Minimizing the differences between measured and simulated particle velocities and front positions allows to identify optimal parameter settings for an observed avalanche event at the Nordkette test site. Using the best-fit parameter values yields deviations below $5-10\%$ for the maximum velocities and the resulting travel lengths. Beyond the best-fit simulations, the applied optimization method shows a wide range of suitable parameter sets causing equifinality within the investigated parameter space. Additionally, the results show that there is a trade-off between the accuracy of an optimization on single observables or the simultaneous optimization of particle and front behaviour. The particle tracking functionalities further allow to investigate the spatio-temporal flow evolution along flow trajectories in a new way. By displaying maximum velocities in dependence of their initial position we reveal that in contrast to the experimental observation, initial position is the determining factor for the maximum velocity along the particles trajectory in the simulations. In conclusion it is possible to identify suitable parameter sets to reproduce the particle motion with high accuracy. However, this analysis also reveals the limitation of the underlying flow model to replicate varying particle properties or corresponding flow regime changes. Analysing AvaNode sensor data also indicates future potential for investigating the influence of snow and particle properties, such as size, shape, or density, on the avalanche flow.



# 1 Introduction

The two major approaches investigating avalanche dynamics are either experimental or computational ones.

So far particle tracking in avalanche dynamics has only gained little attention, reasons are mainly the constrained engineering interest from a macroscopic perspective, coupled with limitations in computational power and available measurement systems (Bartelt et al., 2012). Developments in avalanche modelling, in combination with the availability of low cost sensors and existing measurement technologies built the foundation of investigating avalanche mobility and transport phenomena on

a particle level. In recent years, radar measurements have become the most commonly used technique to capture the dynamic evolution of avalanches (Gauer et al., 2007). These measurements provide a comprehensive overview of avalanches' temporal and spatial evolution, offering a characterization of the flow regime (Köhler et al., 2018). Another experimental technique gaining attention involves in-flow sensors, initially applied in snow chute experiments (Vilajosana et al., 2011) and more recently explored in full-scale rockfall applications (Caviezel et al., 2019). Adapted for snow avalanches in recent years (Winkler

et al., 2018; Neurauter et al., 2023), these AvaNode measurements produce unprecedented datasets on avalanche dynamics at a particle level (Neuhauser et al., 2023). Particle measurements are valuable for exploring particle behaviour in response to environmental influences and terrain. The true significance of particle measurements emerges when they are combined with radar measurements. This synergy allows to determine particle location within the avalanche and their behaviour relative to the avalanche body and in particular the respective flow front.

Tools for the simulation of snow avalanches include a wide range of flow models and numerical implementations (e.g. Christen et al., 2010b; Sampl and Zwinger, 2004; Zugliani and Rosatti, 2021; Li et al., 2021; Hergarten and Robl, 2015; Mergili et al., 2017; Rauter et al., 2018; Oesterle et al., 2022). Their tasks range from simulations for regional avalanche terrain analysis (Toft et al., 2023), to identify endangered terrain for hazard zone mapping or protection forest classification to detailed simulations used for dimensioning mitigation measures. Depending on the application, each model has its own advantages and

disadvantages. For large scale or large area simulations, conceptual data driven models such as Flow-Py (D'Amboise et al., 2022) are used, but also process based, physical models (Issler et al., 2023; Bühler et al., 2018). Classically detailed simulations are performed for operational engineering practice with tools such as RAMMS (Christen et al., 2010b), the former SamosAT (Sampl and Zwinger, 2004) and now AvaFrame (Oesterle et al., 2022); or research questions are investigated in a scientific setting using OpenFOAM (Rauter et al., 2018) or lately for example the MPM method (Li et al., 2021). As the model parameters

of complex avalanche flow models, such as the friction parameters, cannot practically be determined directly by laboratory or field experiments, flow model applications rely on parameter suggestions from guidelines (Gruber and Bartelt, 2007) or parameter optimization through back calculations (Ancey et al., 2003). Within the variety of applied optimization approaches (Eckert et al., 2007; Gauer et al., 2009; Naaim et al., 2013; Fischer et al., 2015) the potential of possible combinations of flow models, friction relations and best fit parameter combinations appears endless, often causing equifinality, which describes the

wealth of valid solutions (Canli et al., 2018; Mergili et al., 2018).

In this work we compare particle measurements to data from numerical simulation particles. There is a fundamental difference between the two: Measurement particles are synthetic particles that should imitate snow granules in the avalanche,





**Table 1.** Summary of the measurement on 2022-02-22 (220222) for the three AvaNodes C07, C09 and C10 with their respective density $\rho$. The first block lists trajectory properties (three dimensional trajectory lengths $\Delta s_{xyz}$ and altitude difference $\Delta Z$), the second block summarizes a first dynamic view on the experiments with maximum velocity $v_{max}$ and flow duration $\Delta t$. A consistent colouring for each AvaNode is used for all visualisations in the following figures and given in the table row *colour*.

| date | Ava Node | colour | $\rho$ [kg/m$^3$] | $\Delta s_{xyz}$ [m] | $\Delta Z$ [m] | $v_{max}$ [m/s] | $\Delta t$ [s] |
|---|---|---|---|---|---|---|---|
| 220222 | C07 | green | 688 | 297.9 | 173.9 | 13.6 | 34.6 |
| 220222 | C09 | red | 415 | 382.1 | 222.5 | 16.3 | 37.1 |
| 220222 | C10 | violet | 415 | 352.8 | 209.4 | 17.1 | 36.9 |

potentially moving relative to the given mountain surface in three-dimensional space. For the utilized simulation approach numerical particles on the other hand represent columns with thickness integrated properties that travel in two-dimensional space along the digital elevation model. Aware of these important differences we analyze a unique data set, with radar measurements and three particle measurements from the same avalanche event. We address the question of the thickness integrated com1DFA flow model's capability to reproduce the measured avalanche front and velocities of measurement particles. To facilitate a comparison with this measurement data, we implement particle tracking functionalities in the thickness integrated gravitational mass flow simulation module com1DFA. We also develop methods to validate and to potentially optimize the particle simulations towards the measurements. The spatio-temporal evolution of the avalanche front and the particle velocities are analysed, addressing the question of the thickness integrated com1DFA flow model's capability to reproduce the measured avalanche front and velocities of measurement particles.

Furthermore, utilizing this particle tracking functionality in com1DFA allows us to investigate the influence of initial and boundary conditions, such as initial position and underlying topography on the resulting maximum velocity or travel length, leading to a better understanding of the similarities and discrepancies between particle behaviour in simulations and measurements. Following this introduction, Section 2 covers the description of the avalanche measurements and the avalanche simulations, introducing the particle tracking features and the approach to evaluate and compare measurement and simulation. The results are described in Sec. 3 and a discussion thereof follows in Sec. 4, with a final conclusion and outlook in Sec. 5.

## 2 Methods for particle and front tracking

### 2.1 Avalanche experiment: AvaNode and radar measurements

The data sets used in this article originates from an avalanche experiment (number #20220025) that was performed on the 22. of February 2022, at the test site Nordkette, Seilbahnrinne, in Austria (Neuhauser et al., 2023). The avalanche was released within avalanche control work after a new snow precipitation event. Some parts of the avalanche reached the catching dam at 1800 m a.s.l. resulting in a maximum altitude difference $\Delta Z$ of 400 m and a projected travel length $\Delta s_{xy}$ of 690 m along the main flow direction.

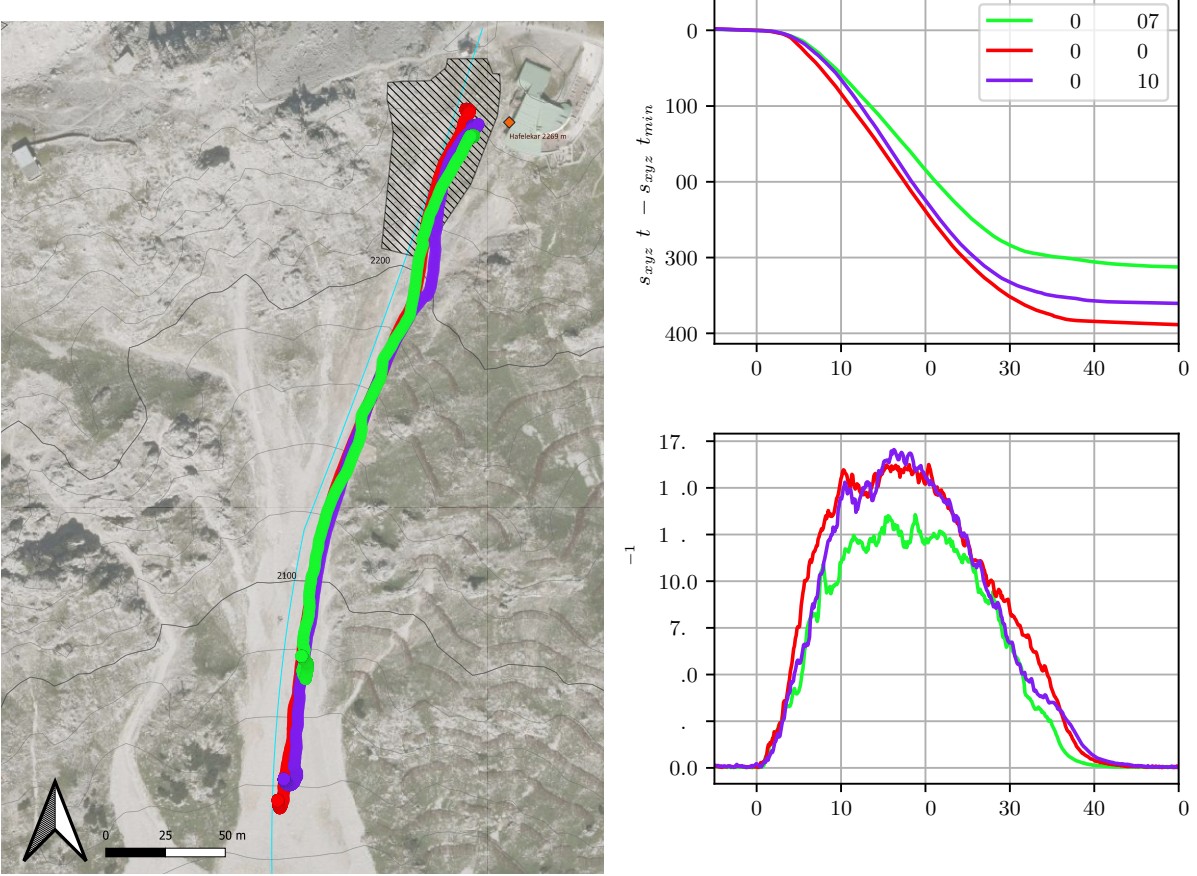

**Figure 1.** Test site Nordkette and experimental data: In the left panel trajectories or GNSS positions of the experiment 220222 are displayed. The dot size of 5 m corresponds to the accuracy stated by the manufacturer. The hatched area represents the release Area. In the right panel the trajectory length $s_{xyz}$ and the temporal velocity evolution of the AvaNodes are shown. $t_{min}$ indicates the starting of the movement and refers to $t = 0$.

The used particle measurement devices are AvaNodes Generation II, with enabled Doppler velocity tracking (Neuhauser et al., 2023). The AvaNodes are equipped with a global navigation satellite system (GNSS) that collects positions and velocities with 10 Hz on three axis. In this experiment three AvaNodes, namely C10, C09 with a density of $415\,\mathrm{kg m^{-3}}$ and AvaNode C07 with a higher density of $688\,\mathrm{kg m^{-3}}$ recorded data. The recorded positions have a horizontal position accuracy of $\sigma_{p,h} = 2.5\,\mathrm{m}$ and the recorded velocities have an accuracy of $\sigma_{v,d} = 0.05\,\mathrm{ms^{-1}}$ on every axis (u-blox, 2022). Tab.1 gives an overview of the collected particle datasets, starting with three-dimensional travel length $\Delta s_{xyz}$, altitude difference between release and deposition $\Delta Z$, maximum velocity $v_{max}$ and duration of overall movement with $\Delta t$ along each particle trajectory. Due to challenging weather conditions the recovery of the AvaNodes took place some days after the experiment, and therefore no information about the burial depths could be observed.
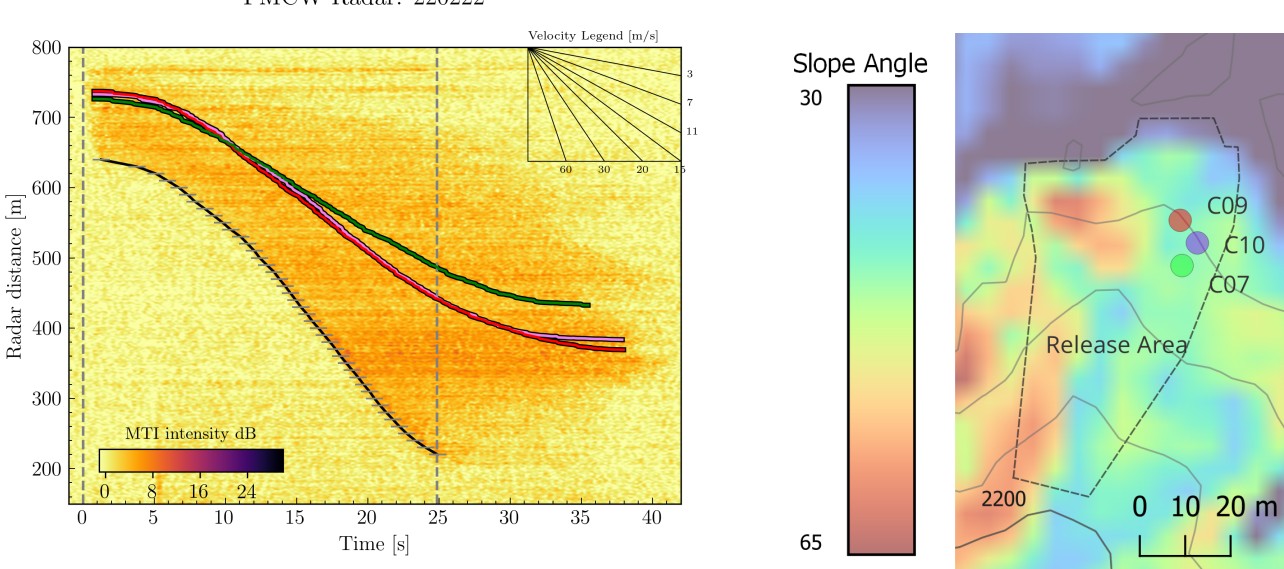

**Figure 2.** Radar measurement and particles distribution in release area. Left: The range-time radar analysis captured on 22nd February 2022 depicts the motion of the avalanche and pinpoints the AvaNode positions in relation to the avalanche front. The black line defines the tracked front, while the coloured lines trace the trajectories of the AvaNodes. Right: Spatial distribution of the AvaNodes in the release area, also highlighting the slope inclination.

The frequency modulated continuous wave (FMCW) radar, the mGEODAR (Köhler et al., 2020) has a spatial resolution of 0.375 m per range gate and a temporal resolution of 50 Hz. In Fig.2 the moving target indication (MTI) plot from the radar measurement is combined with the AvaNode GNSS positions that are transformed into the radar coordinate system and synchronized to the time of release. The radar measurement unfortunately doesn't include the whole run out in the lower part of the test site. Analysis of the radar measurement yields the approach velocity with a maximum value of $26\,\mathrm{m\,s^{-1}}$ in the center part of the avalanche path, where radar line of sight and main flow direction are closely aligned. It also shows the relation of the avalanche front to the AvaNodes, travelling in the middle part of the avalanche approximately 80 m to 100 m behind the front until 15 s, and afterwards separating further to the tail. Additional information on the exact radar location and the evaluation of radar or AvaNode data is provided in (Neuhauser et al., 2023).

### 2.2 Avalanche simulations: AvaFrame com1DFA

AvaFrame is an open-source framework that alongside well-established computational modules, provides various tools for geo data handling, testing, analysis and comparison as well as the possibility to directly add new functionalities. The particle simulations are performed with AvaFrame's computational module com1DFA version 1.7[1] for dense flow avalanches (Oesterle et al., 2022). AvaFrame com1DFA is based on a thickness-integrated flow model, solved by a numerical particle grid method (Tonnel et al., 2023). For this study, particle tracking functionalities are added to com1DFA, allowing to track the numerical

---

[1]https://doi.org/10.5281/zenodo.10033196





particles that start within a predefined radius of a given coordinate point, for example around the initial positions of the AvaNodes. One important aspect for avalanche simulations is the chosen friction relation. For the current study we use an adapted friction relation (Fischer et al., 2015) including a classical Coulomb, Voellmy-like turbulent drag (Voellmy, 1955) and a minimum shear stress term similarly to the effect of snow cohesion (Ligneau et al., 2022), referenced as Voellmy minimum shear stress model:

$$\tau^{(b)} = \tau_0 + \sigma^{(b)} \tan(\delta) + \frac{g}{\xi} \rho_0 \, \bar{u}^2, \tag{1}$$

where $\mu = \tan(\delta)$ refers to the Coulomb friction coefficient, $\xi$ is the turbulent-friction coefficient and $\tau_0$ represents a minimum shear stress. The term with $\xi$ in Eq. 1 increases the friction with increasing velocity and constrains the maximum velocity. The superscript (b) means basal and refers to the shear stress at the bottom surface for $\tau^{(b)}$ and normal stress at the bottom for $\sigma^{(b)}$. To test if we can identify suitable parameter sets within a plausible parameter space, that allow us to reproduce the
measurements, we perform simulations varying $\mu$ between 0.1 and 0.8, with step size 0.1, $\xi$ between 1000 and 10000 m/s$^2$, with step size 1000, and $\tau_0$ between 0 and 150 Pa, with step size 25. Equidistant sampling covers the investigated parameter space and yields 1050 resulting parameter sets. The delineation of the release area is based on the local assessment and performed considering slope angle (between 30 and 60 degrees), terrain curvature and the fracture length of 100-120 m derived from radar measurements (compare Fig.2). The corresponding release thickness is estimated with 70 cm, taking into account local storm board observations in addition to to wind transport into the release area.

### 2.3 Spatial and temporal reference systems

It is necessary to choose a common and appropriate coordinate system when comparing measurements with simulations. Distances derived from radar data are always measured in line of sight towards the radar and therefore one dimensional. For radar measurements it therefore makes sense to keep the comparison in one dimensional space (Fischer et al., 2014).
We follow the approach of Rauter and Köhler (2019), by optimizing the front position between simulation and radar data in the one dimensional radar coordinates. However, com1DFA calculations are based on the thickness integrated governing equations, delivering surface parallel velocities. Particle measurements collect datasets in three dimensional space. Regarding particle simulations and AvaNode measurements it makes sense to keep the overall information about the particle velocity and spatial evolution in three dimensional space. This retains all available information by not transforming it into a one- or
two-dimensional coordinate system.

To achieve time synchronization between the measurement and the simulation, a global synchronization time step, $t_{bomb}$, is used. This time step refers to the release time of the avalanche, defined as 07:52:23 [UTC]. With this interpretation, $t_0$ from the simulation corresponds to $t_{bomb}$ in the datasets from AvaNode and radar.

It is important to point out that AvaNode C07 has a higher density as the two other AvaNodes (C09 & C10) and in particular
compared to the expected snow granule density. The radar measurement lasts for 25 seconds, after which there is no available





information regarding the avalanche front position. This absence of data is attributed to the avalanche front being obscured by an avalanche deflecting dam, causing it to exit the radar's field of view. Consequently, the front comparison is restricted to this specific time frame. The AvaNode measurements on the other side have a duration of roughly 35-40 seconds (compare Table 1) before the movement stops.

**2.4    Particle and front tracking**

Particle tracking is performed with the GNSS modules in the AvaNodes, resulting in positions with 10 Hz temporal resolution and $\sigma_{p,h} = 2.5$ m horizontal position accuracy. The initial position of the AvaNode in the release area and $\sigma_{p,h}$ is used to determine which particles are tracked in the simulation. Each particle in the simulation has its own ID, which is saved throughout the entire simulation. By creating a circle with a radius of 2.5 m around the AvaNode's starting position and identifying the

simulated particle ID's within it. We effectively track around 15 to 20 particles within this radius in the simulation, saving values such as position and velocity at each time step for each particle along its trajectory.

When comparing data from the in-flow sensors (AvaNodes) to the numerical particles of the com1DFA simulations, there are several theoretical differences between the two that hinder a direct comparison and have to be discussed. The AvaNode sensors that flow with the avalanche can be redistributed within the avalanche due to complex flow patterns or due to effects

caused, for example, by differences in the density or size of the snow granules. In com1DFA, numerical particles refer to columns of incompressible snow mass that are forced to travel along the predefined surface. These numerical particles have thickness-integrated properties, such as flow velocity for example and represent the thickness-averaged flow of the avalanche. By comparing the velocity evolution of measurement particles and tracked particles in the simulation, we analyze the thickness-integrated flow model's capability and limitations of replicating the measured particles' behavior.

In order to quantitatively evaluate and compare the simulations to the measured avalanche front (see Fig. 2) we employ a similar tracking method for the simulated avalanche front. Assuming the avalanche moves continuously downhill we track the front by determining the particle with the lowest altitude $z$ value for each time step. In the case of the avalanche front, the tracked particle ID can vary as throughout the simulation, different particles can have the lowest $z$ value at the respective time step.

To ensure that there are enough particles to track at the starting locations of the Avanodes, a dense particle distribution in the release area is needed. The initialization method for the particles in com1DFA was set to mass per particle through number of particles per kernel radius (MPPKR) and the number per particles in kernel radius (nPPK) was set to 50. The default setting results in 458 particles, the adjusted setup used in this study for a more densely distributed particle setup results in 2559 particles. Additional information on the simulation setup, the necessary input data (DEM and release) and the corresponding

parameter setting (configuration files for the com1DFA simulation) can be found in the supplementary material.

GNSS measurements have an output frequency of 10 Hz, the simulations on the other hand allow to export the particle information for a predefined time step. For this analysis, considering computational cost and the effective size of the resulting data set, we compare measurement and simulations on a temporal resolution of 1 Hz.



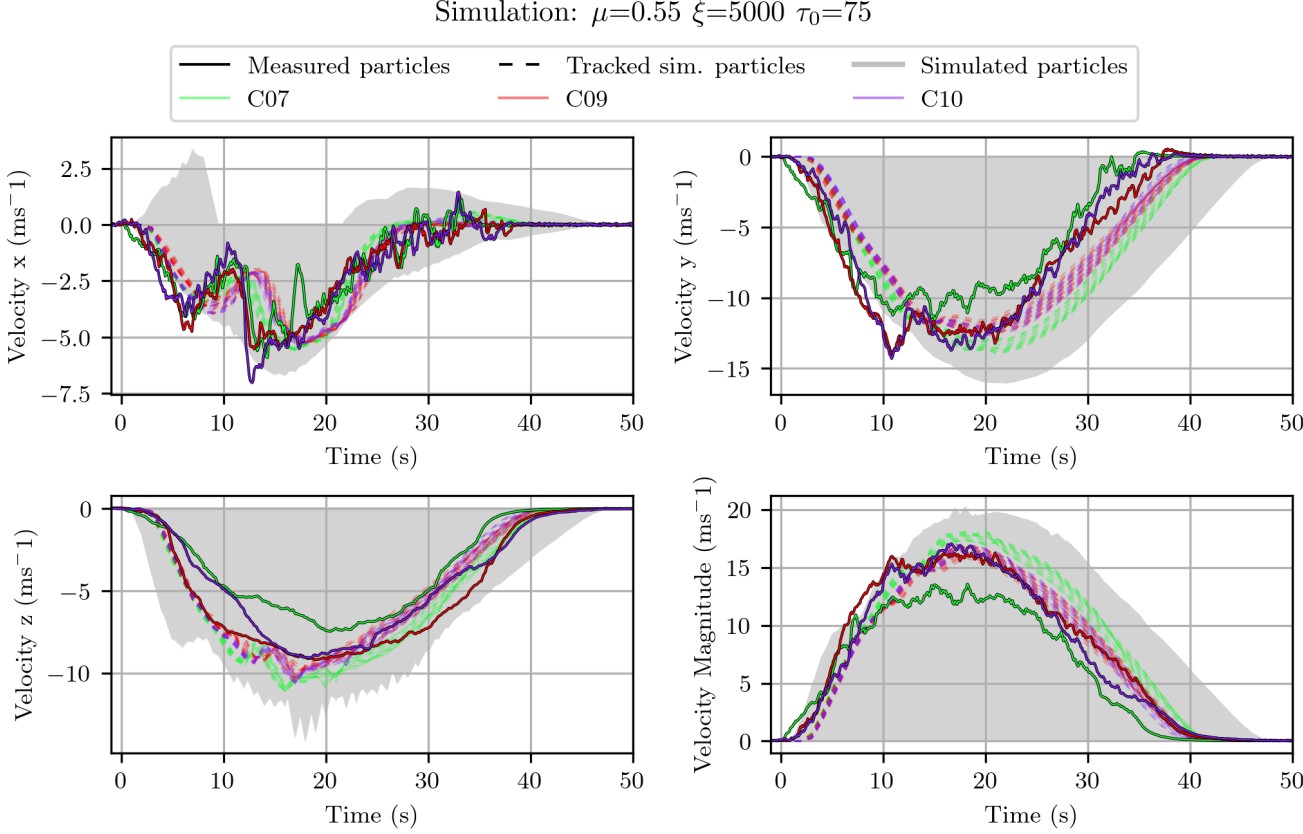

**Figure 3.** Temporal evolution of the velocity along each axis and the corresponding magnitude for the best-fit simulation fitting AvaNode C10 measurements. All simulated particles are visualized in grey, the tracked particles from the simulation are shown as dashed lines and the solid lines represent the measurements.

## 2.5 Particle velocity and front position error

For avalanche simulation optimization the runout or deposition area is mostly used to determine the best-fit model parameters (Sampl and Zwinger, 2004; Christen et al., 2010a; Bühler et al., 2011). These areas rely more on observations made after the event, making them easier to obtain and more numerous compared to inflow measurements taken during an avalanche experiment. Contrarily, our emphasis here is on employing a dynamic evaluation method. This involves comparing range-time data for the avalanche front position and velocity-time data along the individual particle trajectories. When comparing

information regarding the avalanche front position, like its spatial and temporal evolution, the chosen coordinate system is the line of sight towards the radar (Rauter and Köhler, 2019). The AvaNodes record position and velocity in a three dimensional world coordinate system and therefore all further comparisons are done in three dimensions first (Eq. 2) before averaged into a single magnitude (Eq. 3).





When comparing the trajectories one has to keep in mind the obvious differences, with measured velocities representing
free flowing particles in an avalanche, while simulated particle velocities in the com1DFA module of AvaFrame are thickness
averaged properties along the predefined mountain topography.

There are several possibilities to compare the velocities of simulation and measurement. Here we apply a method, evaluating
three dimensional velocities to ensure that the trajectories of the tracked particles are similar. Therefore we calculate the root
mean square error (RMSE) between measurement and simulation, for the tracked particle velocity evolution $\epsilon_v$, on the $x(East)$
axis:

$$\epsilon_{v,x} = \sqrt{\frac{1}{n}\sum_{i=0}^{n}\left(v_{x,i}^{\mathrm{meas}} - v_{x,i}^{\mathrm{sim}}\right)^2}, \tag{2}$$

where $v^{\mathrm{meas}}$ represents the velocity of the AvaNode particle measurement and $v^{\mathrm{sim}}$ the velocity of the tracked simulation
particle assigned to the respective AvaNode for each time step $i$ in the total number of measurement time steps $n$. Analogously
to Eq. 4, values for the particle velocity errors $\epsilon_{v,y}$ and $\epsilon_{v,z}$ are calculated for the y (north) and z (vertical upwards) axis
respectively.

Furthermore the velocity error in all three dimensions, $x, y$ and $z$ can then be summarized in a general or tracked particle
velocity error magnitude $\epsilon_v$, with:

$$\epsilon_v = \sqrt{\epsilon_{v,x}^2 + \epsilon_{v,y}^2 + \epsilon_{v,z}^2} \tag{3}$$

For further analysis the magnitude of the velocity error $\epsilon_v$ is used.

When taking a closer look at the measurement datasets in Fig.3 one can see that there is a slight time delay between the $z$
component and the $x$ and $y$ component. While there is an earlier onset of recorded velocities along $x$ and $y$ axes, velocities also
decrease to zero earlier along these coordinate axes when compared to the $z$ axis. As known from Neuhauser et al. (2023) the
velocity measurements have included Kalman-filters, regarding Fig.3 it looks like the Kalman-filter is strongest on the $z$ value
of the GNSS velocity measurement.

Through particle tracking in the simulations and linking them towards the initial position of the AvaNode, both particles,
simulated and measured one, have the same or at least similar underlying topographic features along their trajectories within
the release area.

For the spatial deviation between experimental and computational results of the front tracking we introduce the position
RMSE of the tracked front evolution $\epsilon_p$:

$$\epsilon_p = \sqrt{\frac{1}{n}\sum_{i=0}^{n}\left(d_i^{\mathrm{meas}} - d_i^{\mathrm{sim}}\right)^2}, \tag{4}$$


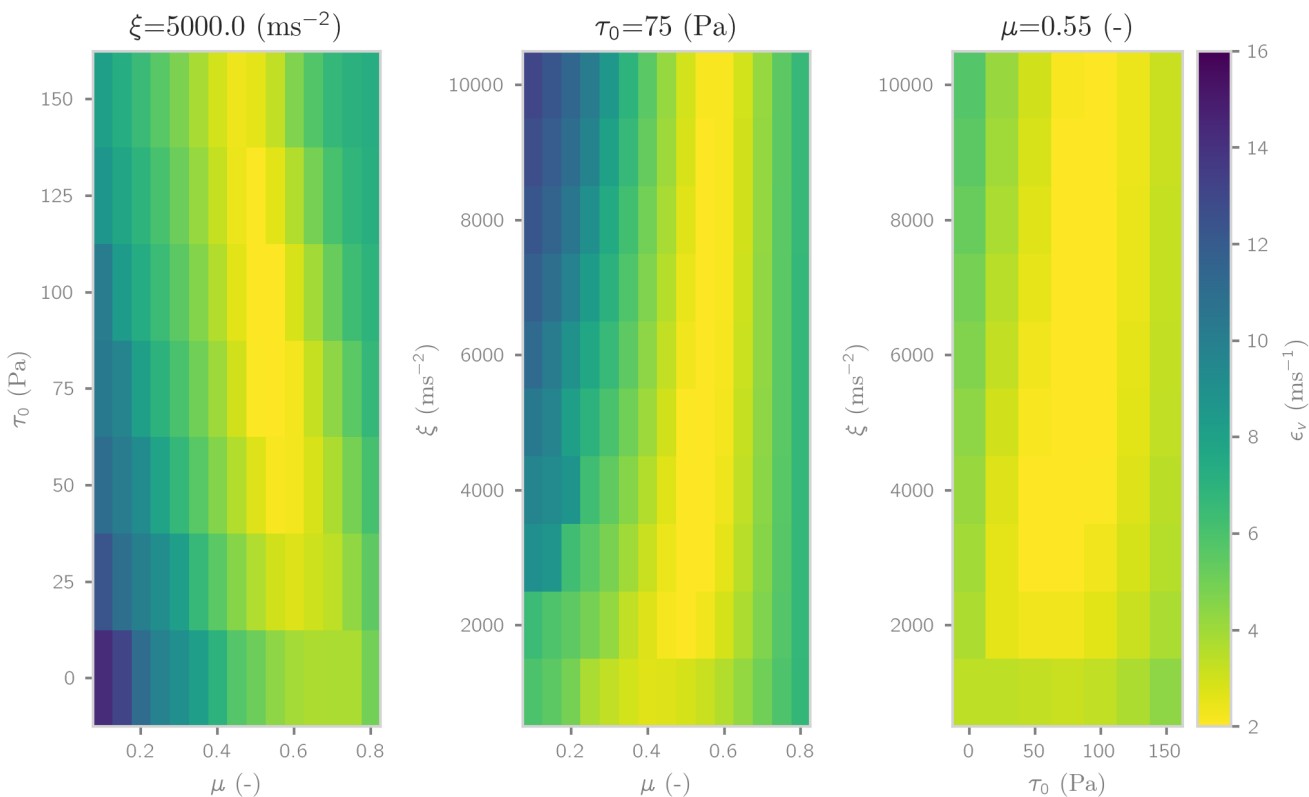

**Figure 4.** Velocity error magnitude $\epsilon_v$ for AvaNode C10 averaged over the corresponding, tracked simulation particles, ranging from 2 to $14\,\mathrm{ms}^{-1}$ in the parameter space around the best-fit parameters.

where $d^{\mathrm{meas}}$ represents the front distance to radar in the measurement, and $d^{\mathrm{sim}}$ the corresponding distance of the simulated particle with the smallest radar range for each time step $i$ in the total number of measurement time steps $n$.

## 3 Evaluating measurements and simulation results

The parameter variation from Section 2.2 results in 1050 avalanche simulations. The number of tracked numerical particles that are initially within the radius of the horizontal positional accuracy of the AvaNodes C07, C09 and C10 are 18,19 and 16 respectively.

### 3.1 Tracked particle velocities

For each simulation the velocity errors $\epsilon_{v,x}$, $\epsilon_{v,y}$ and $\epsilon_{v,z}$ are calculated between the measured particle and the tracked simulated ones, they are then combined to the velocity error magnitude $\epsilon_v$ (Eq 3). This results in multiple $\epsilon_v$ values for every tracked particle. For further analysis and model interpretation we used the mean value of $\epsilon_v$ for all tracked particles, in one simulation,





assigned to the respective AvaNodes. In Fig. 4 the variation of $\epsilon_v$ for AvaNode C10 is displayed, ranging between 1.7 and 14.3 ms$^{-1}$, with one parameter always held constant at the best-fit simulation parameter set. Interestingly one can identify the equifinality with a narrow band of parameter combinations that provide solutions with similar velocity error magnitude for all single parameters, almost covering the entire parameter space. The parameter set with the lowest mean $\epsilon_v$ value for

AvaNode C10 is $\mu = 0.55$, $\xi = 5000\,\text{ms}^{-2}$, and $\tau_0 = 75\,\text{Pa}$, with $\epsilon_v = 1.74\,\text{m/s}$. This parameter set is henceforth referenced as the best-fit for AvaNode C10. The parameter set with the lowest mean $\epsilon_v$ value for AvaNode C09 is $\mu = 0.5$, $\xi = 3000\,\text{ms}^{-2}$, and $\tau_0 = 75\,\text{Pa}$, with $\epsilon_v = 1.77\,\text{m/s}$ and for AvaNode C07 $\mu = 0.5$, $\xi = 3000\,\text{ms}^{-2}$, and $\tau_0 = 150\,\text{Pa}$, with $\epsilon_v = 1.64\,\text{m/s}$.

### 3.2  Tracked front positions

Fig. 5 is similar to Fig.4, but shows the variation of the front position error $\epsilon_p$ (Eq 4), again with one parameter held constant on

the best-fit simulation parameter set. The position error values reach from $\epsilon_p = 4.34\,\text{m}$ for the best-fit to $\epsilon_p = 213.4\,\text{m}$ for the worst-fit simulation. The best-fit parameter for the front is $\mu = 0.4$, $\xi = 5000\,\text{ms}^{-2}$ and $\tau_0 = 125\,\text{Pa}$, resulting in a minimum position error RMSE value of $\epsilon_p = 4.34\,\text{m}$. Technically this is the best-fit, but considering RMSE values lower then $\epsilon_p < 20\,\text{m}$, Fig. 5 indicates a band with possible parameters (light yellow) causing equifinality.

The left panel of Fig. 6 shows the spatial evolution of the avalanche front distance towards the radar derived from the

radar measurements (black solid), extracted from the best-fit front simulation (black dashed) and from the best-fit AvaNode C10 simulation (black dash-dotted). The extracted fronts from all simulations are coloured with the corresponding $\epsilon_p$ value. Since the avalanche exits the radar's field of view after 25 seconds, only this time period is used for the RMSE analysis, nevertheless this is the most dynamic part in the avalanche track including the acceleration state in the steepest part along on the avalanche track. We also analysed the total altitude difference versus the maximum velocity for all simulations, coloured

with the corresponding $\epsilon_p$ value shown in the right panel of Fig 6. The x marks the radar measurement, the circle indicates the best-fit front simulation and the triangle the best-fit simulation for AvaNode C10. The dashed line indicates the maximum velocity an avalanche can obtain according to its altitude difference, following the suggestion of McClung and Schaerer (2006) and Gauer (2014). The relation of maximum velocity and altitude differents originates from observations of many avalanches, however, our simulations show a different behaviour, potentially indicating the influence of the local topography. The grey

dashed-dotted line represents the upper and lower catching dam in the avalanche track. For both catching dams we find many simulations that come to rest, particularly at the upper dam, where the dilute parts of the avalanche stopped and simulation results with low position errors $\epsilon_p$ accumulate around the measured radar front. Both plots give an overview of all simulations that are performed and how well they compare to the radar front measurement.

### 3.3  Comparison of particle and front tracking

The analysis of the best-fit parameters shows rather similar results for the three tracked particles and larger differences when compared to the best-fit front parameters. In order to analyze these differences we focus on one of the particle simulations (AvaNode C10) and compare the simulations results to the best-fit front simulation.


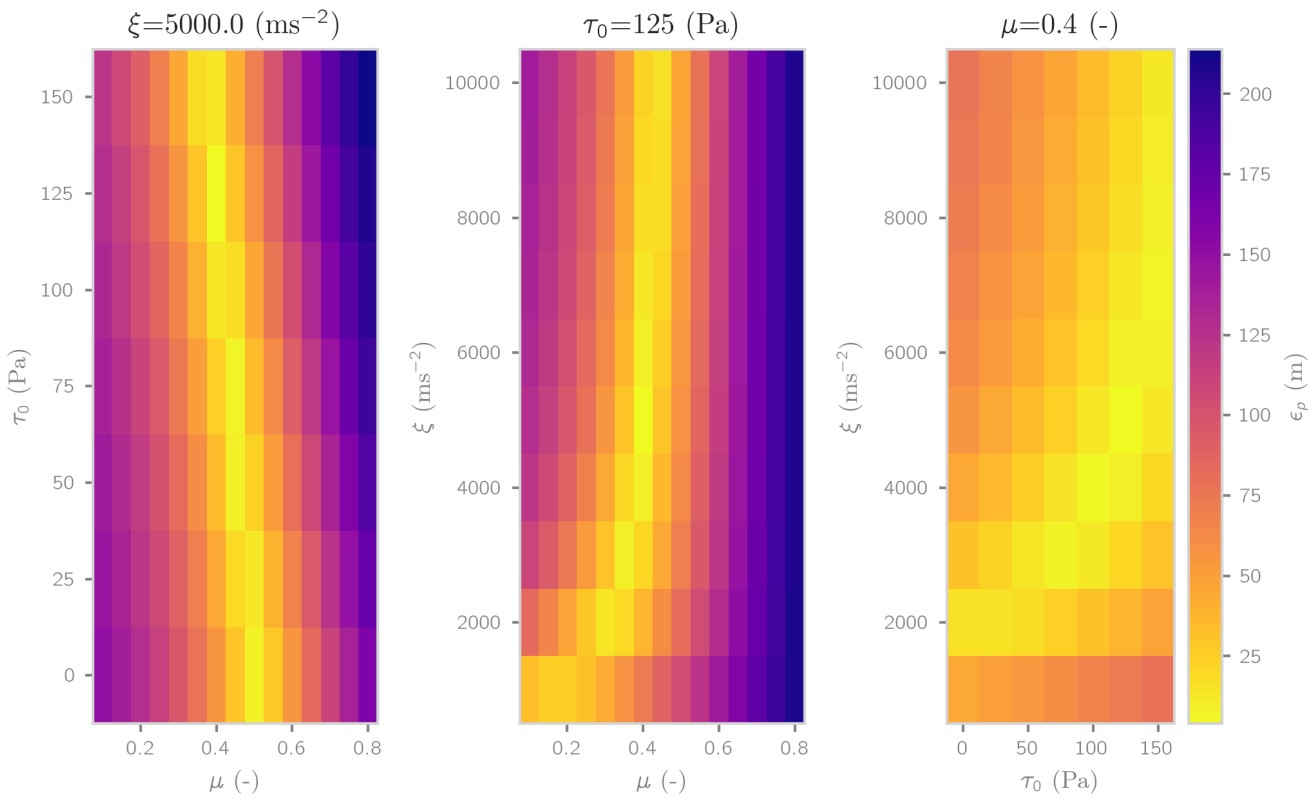

**Figure 5.** Front position error $\epsilon_p$ ranging from 4 to 213 $m$ in the investigated parameter space around the best-fit parameter set.

Fig. 7 shows the temporal evolution of particle velocity (upper panels) and trajectory length (lower panels) for the best-fit simulations fitting the particle measurement data of AvaNode C10 (left) and the recorded radar front (right). All particles of the simulation are visualized in grey as the envelope of mininum and maximum values for all particles at each time step. The tracked particles from the simulation are shown as dashed lines and the solid lines represent the measurements of the AvaNodes. The direct comparison of the two best fit simulations shows that the best-fit front optimization generally yields larger velocities. Although the total duration of the flow is similar in both cases (compare measurement results in Table 1 and Fig. 1), higher velocities and correspondingly longer travel length beyond 600 $m$ appear in the best-fit front simulation.

Fig. 8 shows the Range-Time diagram of the best-fit particle (left) and front (right) tracking simulation parameters. This visualization is similar to Fig. 2, but is particularly useful because it includes front position and velocity data at the same time, for simulation and measurement. The coloured area represents the maximum velocity of all particles at a given radar distance and time step. The black stars indicate the radar measurements of the front positions in time and the solid lines represent the AvaNode measurements. One has to be aware that the slope of the particle trajectories corresponds to approach velocities in the radar coordinate system, which slightly differ compared to the surface parallel velocities of the simulations and the measurement velocities of the AvaNodes.




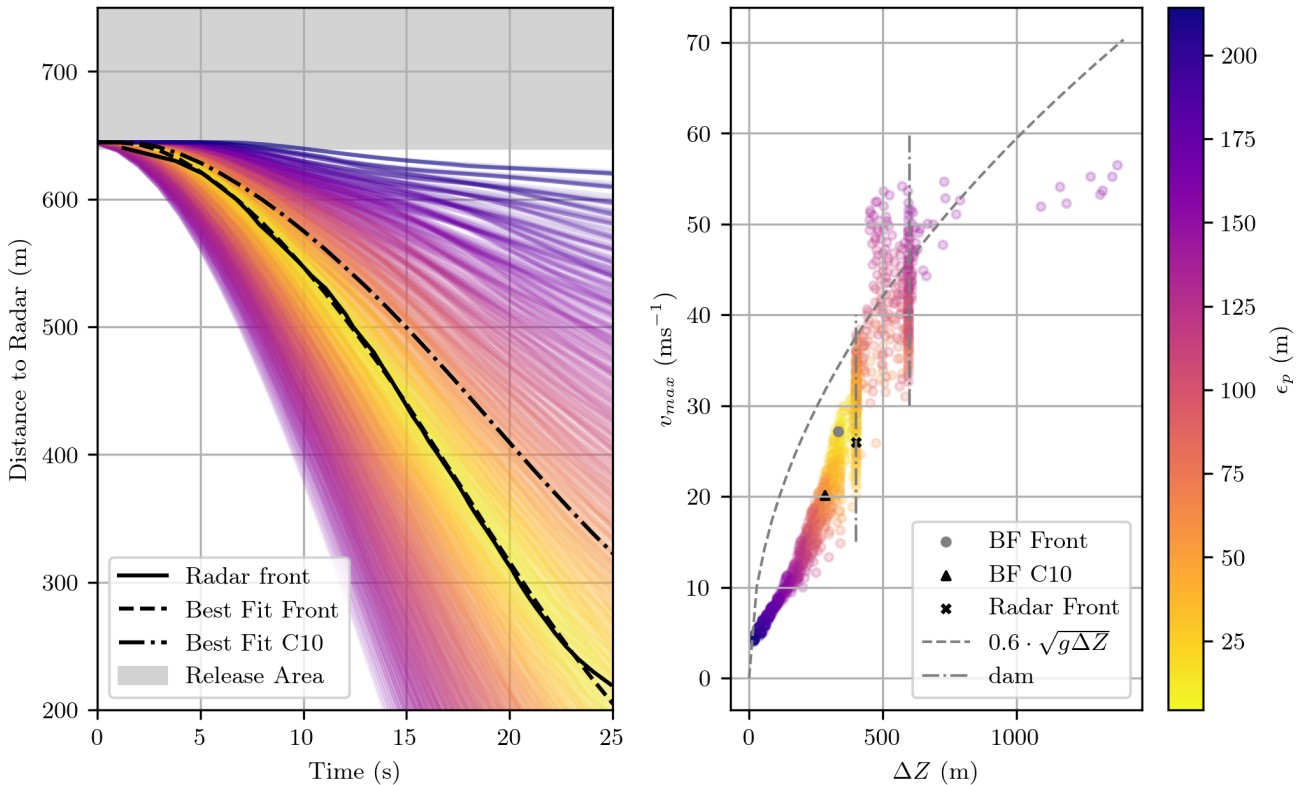

**Figure 6.** The left panel shows the avalanche front positions in relation to the radar, with the corresponding colour from the position RMSE $\epsilon_p$ calculation. The black solid line indicates the radar measurement, the black dashed line the best-fit front simulation and the dashed-dotted line the best-fit simulation for AvaNode C10. The right panel shows the altitude difference versus the maximum velocity of the avalanche front. The $x$ marks the radar measurement, the circle the best-fit front simulation and the triangle the maximum velocity $v_{max}$ and altitude difference $\Delta Z$ of the best-fit simulation for C10.

**Table 2.** Summary of the best-fit parameters for front and AvaNodes, with the corresponding RMSE values. The minimum RMSE value resulting for the corresponding measurement and simulation is highlighted in boldface.

|  | $\mu$ | $\xi$ | $\tau_0$ | Front $\epsilon_p$ | C07 $\epsilon_v$ | C09 $\epsilon_v$ | C10 $\epsilon_v$ |
|---|---|---|---|---|---|---|---|
|  | [] | [ms$^{-2}$] | [Pa] | [m] | [ms$^{-1}$] | [ms$^{-1}$] | [ms$^{-1}$] |
| Front | 0.4 | 5000 | 125 | **4.34** | 4.47 | 3.56 | 3.24 |
| C07 | 0.5 | 3000 | 150 | 81.6 | **1.64** | 3.16 | 2.73 |
| C09 | 0.5 | 3000 | 75 | 50.37 | 2.9 | **1.77** | 1.82 |
| C10 | 0.55 | 5000 | 75 | 63.57 | 2.82 | 1.80 | **1.74** |

The comparison of the two best fit simulations highlights the differences in velocities which also explain the rather large range of the front position error towards the end of the avalanche movement. Similarly to the corresponding measurement (Fig. 2) one can see that the AvaNodes tend to be transported in the main body and towards the tail of the avalanche.


**Figure 7.** Shown are the temporal evolution of the measurement and simulation particle velocities (upper panel) and trajectory length (lower panel). Two best-fit simulations are presented: Best-fit simulation for particle velocities of AvaNode C10 with $\epsilon_v = 1.74$ m/s (left) and the best-fit for the radar front (right, $\epsilon_p = 4.34$ m). All particles are visualized in grey, the tracked particles from the simulation are shown as dashed and the solid lines represent the measurements.

Table 2 summarizes the parameter sets and resulting velocity and position errors for the best-fit simulations, matching the corresponding measurement data. Frontal position and mean particle velocity error ($\epsilon_p$,$\epsilon_v$) are highlighted for each best-fit simulation. The magnitude of the errors reflects the similarity of the best-fit simulations, particularly comparing the the different AvaNode measurements. The front optimization leads to a slightly different parameter result, along with lower position errors and slightly larger velocity error magnitudes. However it is important to note that both, the front position and particle velocity error still remains in a medium range (50 m$\leq \epsilon_p \leq$82 m, $3.24$ ms$^{-1} \leq \epsilon_p \leq 4.47$ ms$^{-1}$ ) for their non best-fit simulations. This indicates that suitable solutions are identified, albeit with reduced accuracy. In Fig. 4, 5 and 6 one can identify the corresponding equifinality bands of parameter sets that lead to low RMSE values. The area for low RMSE solutions is wider for the particle fit in Fig.5 than for the front fit in Fig.4. This indicated that there are more parameter sets with low RMSE values when



Natural Hazards
and Earth System
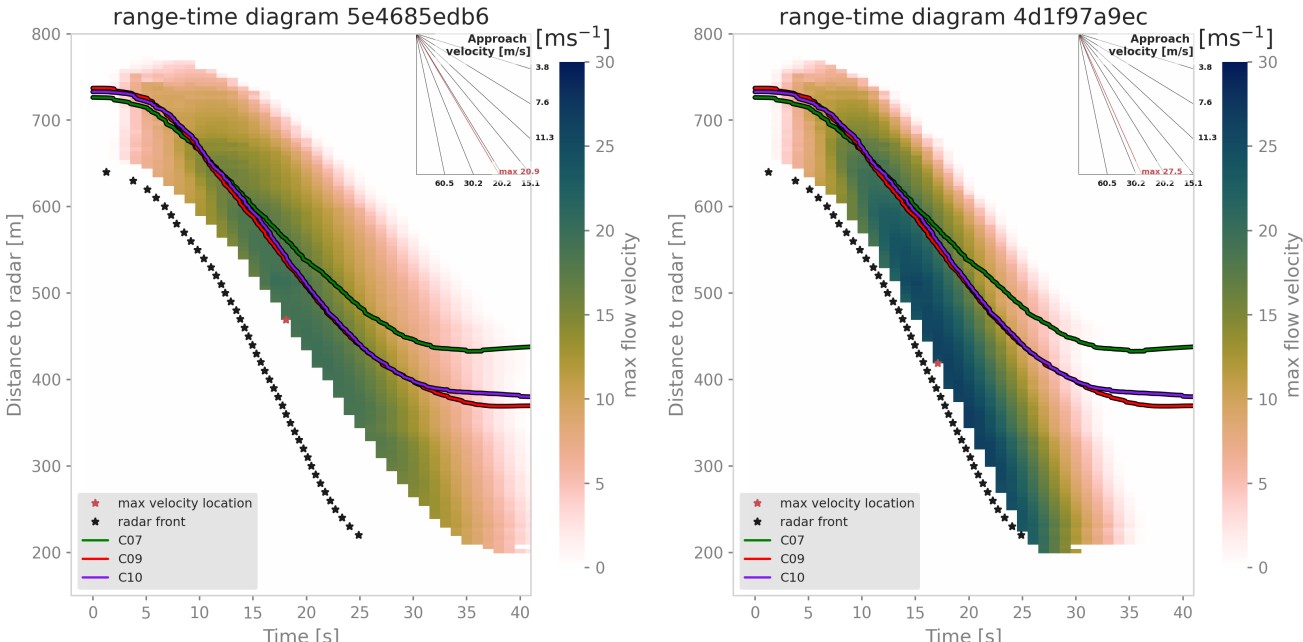

**Figure 8.** Range-time diagram of the best-fit C10 (left) and front (right) simulations, showing the temporal evolution of the maximum flow velocities with distance to the radar. The black stars show the measured the avalanche front, the coloured lines show the AvaNode measurements. The corresponding simulation animation in three dimensional terrain can be found in the supplementary material.

particle velocities are optimized, compared to when front positions are optimized. With the presented method one can now use
275 those bands and search for overlapping areas between best-fit particle and front simulations, with low RMSE values for both
simulations. There are many ways of combining these two parameter ranges.

Table 3 summarizes selected simulation results corresponding to Figs. 7 and 8. The respective measurement data is high-
lighted in Fig. 1 and summarized in Table 1.

Comparing computational and experimental data allows us to deduce the accuracy of the best fit simulations. The resulting
280 differences between maximum particle velocities between measurements (Tab. 1) and simulations (Tab. 3) range between
1.0 m/s and 1.3 m/s, which is less than 10 % of the maximum velocity values. Additionally it is possible to evaluate the travel
lengths ($\Delta S_{xyz}$) or related altitude differences ($\Delta Z$) along their individual trajectories, with the simulated resulting travel
lengths deviating between 6 m and 18 m from the measurements ($\leq 5\,\%$ of the maximum travel length) and a similar range for
the respective altitude differences. Another quantity to assess the accuracy of the simulations is the maximum of the avalanche
285 front velocity, which is obviously closely related to the alignment of the front positions (compare Fig.6). For the best-fit front
simulation (or lowest front position error $\epsilon_p$) we obtain a maximum front velocity of 27.5 m/s compared to 26 m/s for the
corresponding measurement, which is the same level of accuracy that we observe for the particle velocities. Comparing the
range of result values between the different best-fit simulations it can be seen that parameter sets optimized for other result
variables merely provide a slightly lower accuracy for the result variable under consideration.



**Table 3.** Summary of simulation results of the best fit simulations for the front and each AvaNode with the corresponding values highlighted in boldface. Listed are the maximum front velocities $v_{front}$ and frontal altitude difference $\Delta Z$ ) (Fig.8) as well as maximum velocity $\overline{v}_{max}$, travel length $\overline{s}_{xyz}$ and altitude difference $\overline{\Delta Z}$ (Fig.7) averaged for the corresponding tracked simulation particles and each AvaNode.

| | Front | | C07 tracked | | | C09 tracked | | | C10 tracked | | |
|---|---|---|---|---|---|---|---|---|---|---|---|
| | $v_{front}$ [ms$^{-1}$] | $\Delta Z$ [m] | $\overline{v}_{max}$ [ms$^{-1}$] | $\overline{s}_{xyz}$ [m] | $\overline{\Delta Z}$ [m] | $\overline{v}_{max}$ [ms$^{-1}$] | $\overline{s}_{xyz}$ [m] | $\overline{\Delta Z}$ [m] | $\overline{v}_{max}$ [ms$^{-1}$] | $\overline{s}_{xyz}$ [m] | $\overline{\Delta Z}$ [m] |
| Front | **27.5** | **333.0** | 23.8 | 471.0 | 276.0 | 22.1 | 415.3 | 245.5 | 22.2 | 414.0 | 246.4 |
| C07 | 18.8 | 229.8 | **14.9** | **304.7** | **184.4** | 12.9 | 247.9 | 151.1 | 12.9 | 245.6 | 151.2 |
| C09 | 19.9 | 285.4 | 18.4 | 429.2 | 254.1 | **17.3** | **389.2** | **231.6** | 17.5 | 386.7 | 231.7 |
| C10 | 20.9 | 282.3 | 17.7 | 418.7 | 248.4 | 16.4 | 369.8 | 220.6 | **16.6** | **370.8** | **222.7** |

## 4 Discussion

### 4.1 Measurement data and simulation results

We were able to show that the position RMSE for the avalanche front $\epsilon_p$ and the velocity RMSE for particles $\epsilon_v$ provide valuable information about how accurately the simulations can reproduce the measured dynamic behaviour of an avalanche particle or the front. As seen in Table 2 the minimum values for the velocity error magnitude $\epsilon_v$ are below 10% of the maximum velocity detected in those avalanches, which indicates a strong relation between the simulated and measured particles. The advantage of calculating $\epsilon_v$ in three dimensions before combining them is that one could apply weighting factors depending on the accuracy on the different axes. Since it is known and also visible in Figure 3 that the $z$ component in GNSS measurements is the most inaccurate one, this could be weighted less than the horizontal velocity components. It is also clear that it is possible to optimize parameter sets for all measurements together, whereby solutions are found within the equifinality. However, this increases the error for individual result variables. A compromise must be made here between the number of measurements considered and the acceptable errors for certain result quantities.

Fig. 6 shows the best-fit simulation for the avalanche front compared to all simulations coloured in the corresponding RMSE value. In the left panel one can see the evolution of the avalanche front and how well the model can reproduce it, while the best-fit simulation for C10 has a $\epsilon_p$ value of $63.57\,\mathrm{m}$ we reach $\epsilon_p = 4.34\,\mathrm{m}$ for the best-fit simulation front. The right panel shows the overall altitude difference of the front compared to the maximum velocity in this simulation. As one can see the measurement of the front indicates a longer run-out length, leading to deposition in the dam, while the simulation stops earlier, although the maximum velocity of the best-fit simulation is higher. With the front best-fit we ensure that the avalanche front is reproduced accurately in the first 25 seconds, not covering the avalanche run-out towards the deposition zone at the upper dam.

Fig. 7 shows the two best-fit simulations, based on the comparsion to the AvaNode and radar front, with the tracked particles compared to the measured ones. The left panel shows the best-fit C10 simulation, indicating that the tracked simulation particles reproduce the behaviour of the measured AvaNode C10 with highest accuracy (maximum velocity of 16.6 m/s for the simulated vs. 17.1 m/s for the measured particle, see Tables 1 and 3). It is important to note that the best-fit particle simulations are generally accompanied with lower velocities than the best-fit front ones, which is also reflected in the resulting shorter travel length along the particle trajectories and resulting runout.




Fig. 8 illustrates the evolution of the two best-fit simulations in a Range-Time diagram. In the left panel, the movement of the
AvaNodes is reproduced quite accurately, although the simulated and measured fronts show differing evolutions, as previously
observed (Fig.6). On the right panel the front evolution achieves a very high accuracy with a front position RSME value of
$\epsilon_p = 4.34$ m (compare Table 2), but the simulation particles tend to stop earlier than the AvaNodes.

By considering avalanche motion on a particle level, the presented method provides a deeper understanding of how a chosen
friction model performs within the considered plausible parameter space and results in a parameter set that can reproduce the
measurements with the highest accuracy.

Avalanches can exhibit a multitude of flow regimes which either vary from top to bottom along the track or from front
to tail (Köhler et al., 2018) yielding a potential temporal dependency of the related friction coefficients (Buser and Bartelt,
2009). For the observed avalanche and corresponding particle movement we assume the cold flow regime to be most relevant
and only small parts that are slightly fluidized when reaching the maximum runout. Air intake at the flow front may cause the
avalanche body to develop a fluidized layer on top of a dense-flow layer (Issler and Gauer, 2008) and therefore also explain that
small parts of the avalanche traveled further than the main body. Such processes alter the volumetric mixture of snow grains
with interstitial air compared to the tail of the avalanche where one expects a rather dense granular flow (Bartelt et al., 2012).
Even when the analysed medium sized avalanche did not convert towards a fully developed powder snow avalanche with an
intermittent flow regime (Sovilla et al., 2018), there are different frictional relations for the flow of the front compared to the
tail. Our optimizations for the front with radar and the optimization for the particle in the tail indicate that the particle and front
behaviour of the avalanche, measured by the two different systems cannot be fully reproduced by the underlying flow model
and respective friction relation at the same time. This leads to the interpretation that we expect different frictional behaviours
or fitting parameters for the front compared to the tail.

## 4.2    Initial and boundary conditions

The influence of boundary conditions, such as release thickness, release area or topography on simulation results is well known
(Bühler et al., 2011; Bühler et al., 2018). In this paper with particular focus on particle tracking we investigate the interplay
between topography and initial position within the release area. To do so the com1DFA module of AvaFrame has been extended
with the presented particle tracking. The implementation of these functionalities allows us to project simulation results along
the particle trajectories or forwards and backwards in time. With this one can display and analyze how flow quantities, such
as maximum velocity or travel length, develop along potential particle trajectories. In the future, the methodology could allow
predictions where and how something will be transported if the starting point is known. As a first application we use this
methodology to test whether the interaction of local topography and initial position in the release area determines the resulting
maximum velocities.

Fig. 9 shows the reprojection of the maximum velocity along the particle trajectories back into the release area to the respec-
tive initial particle positions for the two best-fit (C10 and front) simulations. Additionally the initial position of the Avanode
sensors and their corresponding measured maximum velocities (see Table 1) are shown. When comparing the simulated to the
measured velocities we observe that the C10 best fit velocities are generally lower than the ones for the best-fit front. This is in


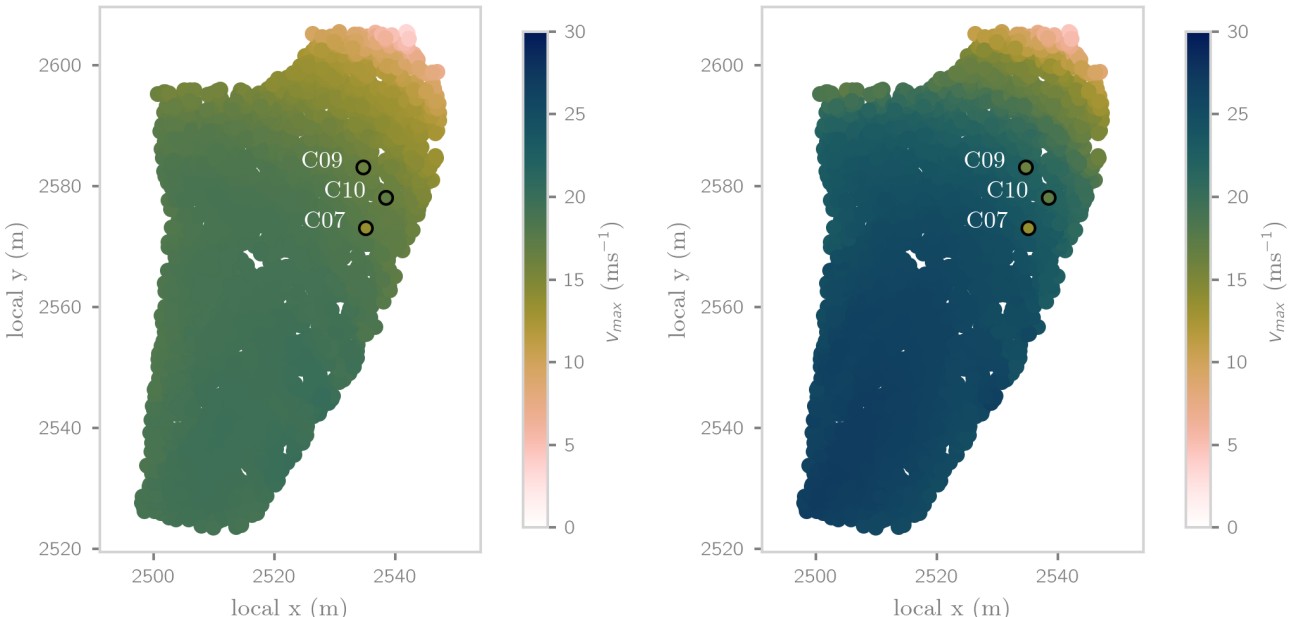

**Figure 9.** This figure illustrates the maximum particle velocity projected to the initial simulation particle position in the release area. The left part represents the best-fit C10 simulation and the right part the best-fit front simulation. The black circles mark the starting points of the AvaNodes, colour-coded based on their corresponding maximum velocity.

line with the previously discussed results. Furthermore the agreement of simulated and measured maximum velocity for C10

and the surrounding simulation particles (with 0.5 m/s difference for the best-fit) highlights the functionality of the particle tracking based optimization. Besides these expected results it is interesting to investigate the similarities between the two best fit simulations. It appears that the determining factor for maximum velocity within the simulations is the relative initial position in the main flow direction within the release area. Particles that travel at the front of the avalanche achieve higher maximum velocities than the ones that follow towards the tail of the avalanche, which may appear counterintuitive considering simple

energy conservation based block model approaches (Körner, 1980), where particles starting at higher altitudes should lead to larger maximum velocities and corresponding runouts. Compared to the relative initial position, the inclination at this position plays a subordinate role, although steeper parts of the release range lead to a higher initial acceleration, which, however, does not lead to the maximum velocities (see slope map in Fig 2 and velocity evolution in Fig. 7 and 8 or the simulation animation in the supplementary material). This is related to the fact that neither particle overtaking nor resorting effects are observed in

the simulation data. At this point, it is important to be aware of the fundamental differences between simulations and measurements. As the simulations are performed using a thickness-integrated model, numerical particles represent two-dimensional columns moving in two dimensional space along the predefined digital elevation model. This involves the effects of pressure gradients in the flow and inhibits overtaking of material at different vertical positions. Due to the thickness-integration and





shallow flow assumption, processes like segregation or overtaking of particles in the vertical dimension cannot be represented
in the simulations.

On the flip side, the measurement data does not confirm this initial position dependency of the maximum velocity and
additionally, the AvaNode C07 (green) is overtaken by the other two AvaNodes. Here we have to remember that the AvaNodes
have different particle properties while all simulation particles have the same generic properties. AvaNode C07 has a higher
density and is initially positioned downhill of the other AvaNodes in the release area, but reaches lower velocities. At this stege,
we cannot infer if the particle property or the initial position are the determining factors for the resulting velocity although the
analysis of other experimental AvaNode data Neuhauser et al. (2023) indicates that particles with higher densities tend to be
differently transported with lower velocities and a different position within the avalanche body.

## 5   Conclusion and outlook

The combination of AvaNode particle and corresponding radar measurements provides a holistic view on the temporal and
spatial evolution of the avalanche. The measurements and simulations provide an unprecedented level of detail with respect
to avalanche particle dynamics. With this study we show how particle and front tracking in experimental and computational
avalanche dynamics help to get a deeper insight into the driving processes behind transport phenomena and mobility as well as
how they are represented in widely applied thickness integrated flow models.

The comparison between measurement and simulation on a particle level delivers new insights on the capabilities and
limitations of the employed model approach. The computational module com1DFA of AvaFrame highlights the potential of
research applications, extending an open source simulation tool for gravitational mass flows that is used for operational hazard
mapping. The implemented particle tracking allows to explore avalanche features such as maximum velocity in new and diverse
ways, namely to project these results along their trajectories to an arbitrary time step, e.g. to their initial position. This approach
provides new insights into the performance of the flow model, the related constraints, and how boundary conditions like particle
properties, topography or initial position influence particle motion during the flow phase. The comparison of measured particle
properties, focusing on their velocity, demonstrates that the underlying model has the capability of reproducing the behaviour
of single measured particles, indicating the comparability of experiment and simulation particles. Using the RMSE velocity
and position method, we replicate avalanche particle and front dynamics quite accurately, although with different parameter
sets. Beyond the best-fit solutions we obtain a wide range of suitable parameter sets within the equifinality. However it is out
of the scope of this paper to address the resulting trade-off between single simulation accuracy and the possibility to replicate
multiple measurements at the same time in more detail. Measurements indicate a dependence of the maximum velocity of
particles on their initial position or particle properties which differs from the dependence found in simulations. This reveals
potential limitations of thickness integrated approaches to reproduce differences between single particle properties like varying
density or processes like overtaking that appear in natural flows through separation and segregation (Gray and Ancey, 2015).
These shortcomings could potentially be resolved by implementing varying particle properties in the simulations, considering
the full three dimensional velocity field (Rauter et al., 2016; Li et al., 2021) or additionally employing multi-phase flow models

(Mergili et al., 2020) with variable frictional and rheological approaches (Jop et al., 2006) or different flow regimes along the path (Bartelt et al., 2012).

In this work, we have taken the first steps towards an in-depth avalanche analysis at the particle level. With more accurate datasets the level of detail could be increased further, providing more insights into avalanche dynamics and the flow regime evolution along the path. Improving the position measurement, i.e. regarding spatial accuracy of the AvaNodes would enhance the overall method. More accurate results would allow to minimize the number of tracked simulation particles, which would be particularly useful in combination with the extension of the spatial distribution and number of measurement particles in the release area. One goal for future investigations and developments would be to get radar measurements of the whole avalanche track combined with more accurate AvaNode GNSS data sets, avalanche run-out distance and deposition area. For the AvaNodes, increasing the accuracy of position measurements, and if feasible, velocity measurements, by using more precise GNSS modules would be beneficial, particularly towards resolving movement in the vertical direction. The focus for radar measurements should be on enhancing the accuracy of the measurements themselves and, if possible, combining two radar types, namely pulse-Doppler and FMCW, to measure the same avalanche. This would provide a better understanding of velocity distribution evolution in avalanches. Additionally, new developments of the AvaNodes could lead to lower densities and a wider range of sensor shape and size, resulting in a more comprehensive analysis of how particle properties influence flow evolution in different locations or parts of the avalanche. This would allow to investigate transport phenomena, such as segregation and separation processes (Gray and Ancey, 2015) or the influence of the vertical resolution of the velocity profile (Tiefenbacher and Kern, 2004; Kern et al., 2009, 2010). Additional sensor systems, such as infrared temperature or pressure sensor in the AvaNodes could deliver new insight to the driving factors of avalanche movement, considering temperature (Vera Valero et al., 2015; Steinkogler et al., 2014; Fischer et al., 2018), related mass evolution through entrainment (Naaim et al., 2013) or to determine the vertical position withing the avalanche.

In conclusion, the advancements in measurement technology and computational modelling pave the way for a deeper and more comprehensive understanding of avalanche dynamics, ultimately enhancing our ability to predict and mitigate avalanche-related hazards, towards predicting flow intensities with respect to their initial position and along flow trajectories, that may serve useful for optimal search design for burials or terrain classification with respect to the potential destructiveness.

*Data availability.* Additionally to this paper we deliver all initalisation files and input data required to reproduce the best-fit simulations. With this dataset comes a readme file that describes how to setup the simulations.

*Author contributions.* M.N. developed the original idea. J.-T.F., W.F. and A.W. contributed to the paper concept. M.N., A.W. and A.K. processed and analyzed the simulation and measurement data. F.O., A.W., J.-T.F and M.N. coordinated and contributed to the simulation tool extension development. J.G., F.D., J.-T.F., A.K. and M.N. lead and contributed to the measurement device developments. M.N. drafted



the manuscript and designed the figures with contributions from the co-authors. W.F., J.-T.F. and F.O. are responsible for supervision of the work. F.D., J.G. and J.-T.F. organized the project funding. All authors discussed the results and commented on the manuscript.

*Competing interests.* The authors declare that they have no known competing financial interests or personal relationships that could have
430   appeared to influence the work reported in this paper.

*Acknowledgements.* This work was conducted as part of the international cooperation project "AvaRange - Particle Tracking in Snow Avalanches" supported by the German Research Foundation (DFG, project No. 421446512) and the Austrian Science Fund (FWF, project No. I 4274-N29). Additional financial support came from the open Avalanche Framework AvaFrame (https://www.avaframe.org/), a cooperation between the Austrian Research Centre for Forests (Bundesforschungszentrum für Wald; BFW) and the Austrian Avalanche and Torrent
435   Service (Wildbach- und Lawinenverbauung; WLV). The authors would like to further acknowledge the contentual and technical support by Engelbert Gleirscher and Jannis Aust, the Nordkette ski resort for their support and indispensable avalanche control work and Lambda4 for providing their innovative, helpful avalanche recovery systems.



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
