# Peer review of "Particle and front tracking in experimental and computational avalanche dynamics"

_Natural Hazards and Earth System Sciences, 2024_

## Referee Comment (RC2)

[revised manuscript text omitted]

* * *
**Page: 2**

**Author: Hans-Peter Marshall    Subject: Comment on Text    Date: 3/9/25, 4:38:27 PM**
Is this one sentence a separate paragraph?

**Author: Hans-Peter Marshall    Subject: Cross-Out    Date: 3/9/25, 4:38:42 PM**

**Author: Hans-Peter Marshall    Subject: Cross-Out    Date: 3/9/25, 4:38:53 PM**

**Author: Hans-Peter Marshall    Subject: Inserted Text    Date: 3/9/25, 4:39:21 PM**
due to

**Author: Hans-Peter Marshall    Subject: Inserted Text    Date: 3/9/25, 4:39:46 PM**
,

**Author: Hans-Peter Marshall    Subject: Inserted Text    Date: 3/9/25, 4:41:12 PM**
scientists

**Author: Hans-Peter Marshall    Subject: Inserted Text    Date: 3/9/25, 4:42:00 PM**
,

**Author: Hans-Peter Marshall    Subject: Inserted Text    Date: 3/9/25, 4:41:50 PM**
identification of

**Author: Hans-Peter Marshall    Subject: Inserted Text    Date: 3/9/25, 4:42:28 PM**
are used

[revised manuscript text omitted]

Page: 6

| | Author: Hans-Peter Marshall | Subject: Cross-Out | Date: 3/9/25, 6:26:09 PM |

| | Author: Hans-Peter Marshall | Subject: Inserted Text | Date: 3/9/25, 6:26:17 PM |
a

| | Author: Hans-Peter Marshall | Subject: Comment on Text | Date: 3/9/25, 6:28:22 PM |
need more info about the radar position and orientation to understand how to obtain this from Fig. 2

| | Author: Hans-Peter Marshall | Subject: Inserted Text | Date: 3/9/25, 6:28:38 PM |
as

| | Author: Hans-Peter Marshall | Subject: Comment on Text | Date: 3/9/25, 6:29:02 PM |
how was this part estimated?

| | Author: Hans-Peter Marshall | Subject: Comment on Text | Date: 3/9/25, 6:30:26 PM |
what is this expected density? How was it estimated?

[revised manuscript text omitted]

---

## Author Comment (AC1)

We are grateful for the suggestions of the reviewer and will revise the manuscript carefully. In the following we give detailed answers (in blue) to the comments.

**General comment**

The paper by Neuhauser et al presents a detailed analysis of the dynamics (front position, « internal » velocities thanks to particle tracking) of one single medium size avalanche event using a combination of approaches that involve full-scale experiments with advanced instrumentation and simulations : i) particle tracking within the avalanche using the AvaNodes' technique recently applied to snow avalanches, ii) radar measurements, and iii) specific depth-averaged simulations based on a Voellmy rheology enriched with cohesion that allow (simulated and specific) particle tracking, namely the com1DFA module of AvaFrame.

I found the paper quite difficult to digest and to follow because many statements that are quite unclear when they appear for the first time are then discussed/solved later at different locations of the paper but it is frustrating that explanations come too late.

That's crucial, but could be improved!

A: We particularly tried to take into account the reviewer's comment, emphasizing the differences between snow granules, AvaNodes and numerical AvaFrame com1DFA particles.

I think that the story can be improved in many places to help the reader. In parallel there is a lack of explanation on some points that may be trivial for the authors but certainly not for the reader. I will try to provide a number of detailed comments about this issue.

More generally, I'm not sure about the key objective of the paper? Do the authors want to use AvaNodes' measurements to better constrain numerical simulations based on a depth-averaged model with a closure Voellmy law with cohesion? Do the authors want to better/test calibrate Avanodes' technique based on radar measurements? Do the authors intend to investigate the granular processes at stake within the bulk of an avalanche flow thanks to particle tracking in both experiments and simulations, having in mind that by construction Avanodes' experiments and com1DFA simulations have their drawbacks and have fundamental differences (see detailed comments)? I think the authors should explain more explicitly their objectives and try to propose a better structure for the paper so that the key objectives become clear.

**A.: The key objectives are:**

(i) developing a general framework for testing and calibrating thickness-integrated models, applied here for example to an extended Voellmy law with cohesion using in flow measurements with AvaNodes and radar;

(ii) implementing particle tracking functionalities in open-source flow model.

We will include a section *Objectives*, that includes the last two paragraphs of the Introduction and a more detailed description of the key objectives. The section would start at Line [56]:

Objectives

This study aims to evaluate and improve the capability of a thickness-integrated flow model to reproduce observed avalanche dynamics by making use of a unique dataset combining radar-based front tracking and in-flow measurements from three synthetic sensor nodes (AvaNodes). A crucial challenge in this context lies in the conceptual difference between measured and simulated particles: while the measured AvaNodes are designed to mimic snow granules moving three-dimensionally with the avalanche flow, the numerical particles in the simulation represent columns with thickness-integrated properties constrained to two-dimensional motion along the digital elevation model.

The first key objective of this work is the development of a general framework for testing and calibrating thickness-integrated flow models, demonstrated here using an extended Voellmy law including cohesive effects, based on high-resolution field data. The second objective is the implementation of particle tracking functionalities within the open-source simulation module com1DFA, enabling a direct comparison between simulated and measured particle trajectories and velocities.

Together, these developments allow for a comprehensive investigation of the spatio-temporal evolution of the avalanche front and particle velocities. In addition, they enable an analysis of how initial and boundary conditions, such as the starting location or underlying topography, affect resulting quantities like maximum velocity and runout distance. Ultimately, this work contributes to a more detailed understanding of how thickness-integrated models can be aligned with real avalanche behaviour through targeted calibration and validation efforts.

Although I have the main concerns above regarding this initial version of the manuscript (see my details comments below) I think that the content of the paper is original and of great interest and I would be happy to read a thoroughly revised version of the manuscript.

**Detailed comments**

C1. line 26. At this stage of the paper, it is not clear which type of particles you are speaking of here: there is a large spectrum of particles in snow avalanches from 'tiny' particles of ice, that are relevant to aerosols (see for instance Rastello et al, J. Glaciol. 2017), to 'much larger' snow aggregates, relevant to dense snow avalanches, either dry or wet (having in mind that wet snow 'granules' are larger than dry snow 'granules')? I would suggest you to say that you deal with dense avalanches and snow granules/aggregates --made of hundred/thousands (if not more) of snow grains, right at the start of the paper (though it is obvious to you I think).

A1. To clarify on what we are focusing on this work we suggest rewriting the first part of the Introduction section and implement a general overview of the particles that can occur in an avalanche and which of those are handled in this paper. It will start at Line [25]:

Snow avalanches are dynamic and complex natural phenomena that involve the movement of large masses of snow down a slope. These masses are composed of a wide range of particles, from tiny ice crystals and aerosols to much larger snow aggregates, each with distinct properties and behaviours. At one end of the spectrum, snow avalanches can contain fine, airborne particles such as ice crystals, which are relevant to studies on aerosols (Rastello et al., 2011). These fine particles are typically associated with the powder cloud of powder snow avalanches. At the other end, avalanches can involve much larger snow aggregates, which are formed by the clumping together of snow grains to larger granules (Bartelt and McArdell, 2009). These granules are particularly relevant to dense snow avalanches, whether dry or wet.

Wet snow granules, for example, are generally larger and heavier compared to the smaller snow granules found in dry avalanches (Steinkogler et al. 2015).

In this study, we focus on dense flow avalanches, which are characterized by the movement of snow granules made up of hundreds or even thousands of individual snow grains. These clumps exhibit different dynamics to individual ice grains and are of particular interest due to their potential for high impact pressures and larger scale flow behaviour (Sovilla et al. 2018). The observation and study of such dense flow avalanches, which involve a significant interaction between snow granules, is crucial for understanding avalanche dynamics and improving predictive models.

C2. Table 1; and lines 83-84. The density of the Avanodes: why choosing such densities of 688 kg/m3 and 415 kg/m3? I would expect more explanation. It seems obvious that the trajectories of isolated particles are influenced by their density. Moreover, you don't give the size of Avanodes, which is another important input. Could you please elaborate more on this? It has something to do with the following comment (C3).

A2. We added a more detailed explanation on the size of the AvaNodes and why we came up with these densities. The used densities are inherent to the prototype design, with efforts focused on minimizing weight while accommodating necessary hardware, power source and casing constraints. A density of 415 kg/m3 represents the lowest achievable value within the current design limitations. With the gained experience and further advancements in using these sensors in avalanche experiments, it is now possible to produce a wider range of densities and likely reduce the minimum density of the AvaNodes.

We will add a paragraph to make this clear, at Line [84]:

While snow particles in avalanches tend to have a density between 100 and 400 kg/m3 during the movement and 250 to 400 kg/m3in the deposition (Dent et al., 1998), it was the goal to achieve similar densities for the AvaNodes. However, due to design constraints, the minimum achievable density was 415 kg/m3.

The AvaNodes are cubelike bodies with an outer length of 16 cm, which is comparable to the typical size of snow granules found in the deposition zone (Bartelt and McArdell, 2009). The inclusion of a higher-density node (688 kg/m3) was intended to introduce variation, facilitating the investigation of potential differences in the behaviour of varying densities during flow.

C3. lines 62-63. Not easy... as there is one more degree of freedom in reality compared to the depth-averaged simulations used, there are potential errors. For instance, segregation processes, mixing, and the existence of secondary flows in granular flowing media is well established and we may expect strong differences between trajectories of single snow granules in reality and synthetic particles in a depth-averaged simulation framework that do not take into account all this complexity.

A3. Indeed, this is a crucial point, however all the mentioned processes cannot be modelled by a thickness-integrated model. As a first step it is important to include the functionalities into the simple model which can then be extended to more sophisticated models.

C4. lines 70-71. There are obvious/fundamental differences between particle tracking from Avanodes' experimental technique and particle tracking in the simulations: i) the particle are

not the same, ii) the Avanodes have a size and a density (and a shape), are mixed with a snow granular assembly and subject to a number of complex granular processes (size and density segregation, geometrical trapping, mixing, secondary flows, etc.) that are not taken into in the simulations. As such trajectories shouldn't be the same in the end. I think you should detail and explain these crucial differences earlier in the text and then explain why the cross-comparison still remains relevant? And what is your key objective of that cross-comparison? Beyond the errors inherent to each approach (deciphering the true trajectories of the snow granules is challenging), do you expect little or huge gap between the measured and simulated velocities and trajectories?

A4. Thank you for this valuable comment. We have included a discussion of the fundamental differences between real snow particles, AvaNodes, and the numerical particles used in the simulations earlier in the introduction to clarify the limitations and motivation for this comparison. Snow avalanches consist of complex granular flows with particles varying in size, shape, and density, and are subject to mechanisms such as segregation, mixing, and secondary flow structures. The AvaNodes, although designed to approximate the behavior of snow granules, are synthetic objects with fixed shape, size (16 cm cube), and density, which inevitably influence their dynamics through interaction with the real snow granules. In contrast, the simulation framework used in this study, based on thickness-integrated flow models, represents the avalanche body as depth-averaged columns, which do not resolve individual particle interactions.

Despite these discrepancies, we consider the cross-comparison meaningful and relevant for two main reasons. First, most operational avalanche models do not provide individual trajectories of particles at all. Therfore, implementing and comparing particle tracking, even under simplifications, represents a novel step toward bridging the gap between granular-scale measurements and continuum modeling. Second, while we do not expect trajectories and velocities of AvaNodes and simulation particles to match precisely due to the outlined physical differences, our objective is to assess whether the general spatio-temporal trends can be reproduced, such as relative timing, velocity ranges, and travel distances. This enables a first-order validation of model behavior with respect to real in-flow observations and opens the door to further refinement of simulation frameworks with more granular realism.

We will change and add a paragraph at Line 56:

Snow granules within avalanches undergo a range of complex processes, including segregation by size and density, mixing, and secondary flow structures (Edwars et al 2022). Additionally, the granules themselves evolve over time through aggregation and crushing, directly affecting flow dynamics (Marks and Einav, 2015, 2017). These mechanisms play a central role in shaping avalanche behaviour and internal structure yet are often difficult to observe directly. Understanding these granular interactions is essential for interpreting particle-level measurements and their implications for avalanche dynamics.

In this context, it is important to acknowledge the fundamental differences between natural snow granules, the synthetic sensor particles (AvaNodes), and the numerical particles used in thickness-integrated models such as com1DFA. The AvaNodes are rigid, cubic objects with fixed size and density that interact with the granular snow medium, potentially undergoing effects such as geometrical trapping, density-driven segregation, and complex mixing processes. In contrast, the numerical particles represent depth-averaged flow columns and do not capture these microscale interactions, with an artificial numerical size that does not directly relate to a physical scale. Despite these inherent limitations, the implementation of

particle tracking in the simulation allows for a first-order comparison between observed and simulated particle behaviour. This enables an evaluation of whether such models can reproduce key trends in avalanche dynamics, such as the timing, spatial evolution, and magnitude of particle velocities, even without explicitly resolving granular physics. Such comparisons are an important first step toward enhancing model realism and integrating inflow sensor data into model validation frameworks.

**C5. lines 94-95: why? Please explain here or refer to the explanation that will come later (see comment C10)**

A5. We will include the explanation for Question C10 or Line 136-137.

The radar measurement unfortunately doesn't include the whole run out in the lower part of the test site, because the avalanche is being obscured by an avalanche deflection dam in this area, causing it to exit the radars field of view.

C6. line 105. Why 'a minimum'? Could you please elaborate. I think you just implement a constant yield stress that may refer to a 'cohesion' yield stress and add it to the total Voellmy stress.

A6. This is correct. Minimum shear stress refers to the fact that the material needs to exceed this stress limit to initiate movement. Conceptually it can be compared to cohesion and is a part of the total stress s - we choose to stay in line with the wording of previous publications (Sampl and Zwinger 2004, Fischer 2013).

To clarify, we adapted the text accordingly [Line 106-109]:

For the current study we use an adapted friction relation (Fischer, 2013), referenced as Voellmy minimum shear stress model, including a classical Coulomb, Voellmy-like turbulent drag (Voellmy, 1955) and a shear stress limit, termed minimum shear stress (Sampl and Zwinger 2004). The minimum shear stress models the effect of snow cohesion (Ligneau et al., 2022), which needs to be exceeded to initiate movement.

**C7. line 115: why such a huge upper value for the turbulent friction?**

A7. As these are the results of the first iteration, we wanted to choose the parameter ranges as widespread as possible, to catch all possible outcomes (Fischer et al., 2015). By using a value this high, we wanted to look at the effect, when the turbulent friction has nearly no impact on the simulation. In a second iteration one could now use tighter parameter ranges, to focus more on the important areas. But we first needed to identify those.

We will add a paragraph at Line [117]:

As this study represents the first iteration, the parameter ranges were deliberately chosen to be as broad as possible to capture the full spectrum of potential outcomes (Fischer et al., 2015). In a subsequent iteration, narrower parameter ranges could be employed to focus on the most relevant regions identified in this initial exploration.

C8. line 117: the range of 'cohesion' you use is small, finally (when compared to a typical frictional stress: mu (rho g h (easily more than 1000 Pa based on h = 1 m, density of 300

**kg/m3 and mu=0.5). Why not using a classical Voellmy law without cohesion? Could you please elaborate more on this point?**

A8. This is a good point. The used range for the minimum shear stress and the found optimum for this parameter seems to be small compared to the frictional resistance. However, even these small values influence the flow dynamics that much, that the optimal parameter combinations contain a minimum shear stress considerable larger than zero, which would not be the case for a negligible minimum shear stress. We assume, that the influence of the minimum shear stress is larger for the small avalanches studied here as for large avalanches, as the friction resistance is lower for small flow thicknesses than for larger ones.

We will add a paragraph following the last paragraph of A7:

Within this process, we also evaluated the performance of the classical Voellmy friction law. The results showed that this simpler model did not achieve fits of comparable quality to those obtained with the extended Voellmy model including a minimum shear stress term. Although the absolute values of the minimum shear stress remain relatively small when compared to typical frictional stresses in dense flows, the optimization did not converge toward a cohesion value of zero. This indicates that the presence of a cohesive component, even if minor in magnitude, enhances the model's ability to replicate the observed avalanche dynamics. The findings thus support the use of an extended friction formulation to better represent the physical processes within dense snow avalanche flows.

C9. lines 134-135: yes, this is an important (crucial) point. Could you elaborate more on the expected density of snow granules versus the one of AvaNodes? Did you have any measurements of snow granules after the event. Or at least an indication of the type of snow (dry, wet) ? Typical size of the snow granules? I would suggest that you give more information on the survey of the specific avalanche event investigated, right at the start here. Some information is provided a bit later (see around line 325) but it is not enough I think.

A9. Unfortunately, no direct measurements of snow granules were conducted after the event. However, based on meteorological records and observations from the deployment team, the avalanche can be classified as a dry, dense flow avalanche.

We will expand paragraph Line 75 -80 with more information about the avalanche event:

The data sets used in this article originate from an avalanche experiment (number #20220025) that was performed on the 22 of February 2022, at the test site Nordkette, Seilbahnrinne, in Austria. The avalanche was released during avalanche control work after a new snow precipitation event of around 40 cm new snow at Seegrube. Some parts of the avalanche reached the catching dam at 1800 m asl resulting in a maximum altitude difference  $\Delta Z$  of 400 m and a projected travel length  $\Delta s_x y$  of 690 m along the main flow direction. More details to this avalanche event is found in (Neuhauser et. al, 2023).

According to the international avalanche classification (avalanche atlas (UNESCO, 1981)), the observed avalanche classifies as A2B1C1D2E2F4G1H1J4, corresponding to: slab avalanches (A2), with a sliding surface within the snow cover (B1) and dry snow (C1) in the zone of origin, channelled avalanches (D2), dominated by the dense, flowing part (E2) in the zone of transition and mostly fine (F4), dry (G1), clean (H1) deposits in the zone of deposition. The avalanche has been artificially triggered within avalanche control work (J4).

C10. lines 136-137. This explanation should come earlier (see comment C5).

**A10. We will change this accordingly.**

**C11. lines 147-148. Yes, this should be emphasized earlier / clearly stated I think to help the reader to follow.**

A11. Thank you for this valuable suggestion. In response, we refer to our Answer A4, where we have implemented an additional paragraph in the *Introduction* section that clearly outlines the fundamental differences between natural snow granules, the AvaNodes used in our experiments, and the numerical particles in the simulation framework. By addressing these differences at the beginning of the manuscript, we aim to provide readers with the necessary context to better understand the limitations and scope of the comparative analysis.

C12. lines 149-150. Yes, and much more than that: grain size segregation, grain density segregation, mixing, and secondary flows, and even changes over time of the snow granules as a direct effect of competitive aggregation and crushing of snow granules. I think you should elaborate more on the complexity of granular processes involved in snow avalanches and refer to key papers about those granular processes. I have for instance in mind the recent work by Marks and Einav (Geophys Res Lett 2015, Granular Matter 2017). This is important to highlight/state the gaps between the true snow granules' dynamics and the one from AvaNodes. I don't even speak of the gap between AvaNodes and simulated depth-averaged particles.

A12. We try to answer this question in first part of A4, which we will insert in the Introduction at Line [56]:

Snow granules within avalanches undergo a range of complex processes, including segregation by size and density, mixing, and secondary flow structures. Additionally, the granules themselves evolve over time through aggregation and crushing, directly affecting flow dynamics (Marks and Einav, 2015, 2017). These mechanisms play a central role in shaping avalanche behaviour and internal structure yet are often difficult to observe directly. Understanding these granular interactions is essential for interpreting particle-level measurements and their implications for avalanche dynamics.

C13. lines 160-165: OK you are speaking of particle distribution in your simulations. But what about the size of your depth-averaged particles? How does it compare to the typical size of snow granules and/or the size of Avanodes? Does it make sense to compare this? Could you discuss more on this?

A13. Numerical particles do not have a size. They have a mass and a volume and therefore an artificial numerical size can be computed, still this is not comparable with a real size of an avalanche particle. It is by no means a particle because of the thickness-integration process.

The problems and limitations when comparing the AvaNodes to numerical particles, are discussed in lines 147-154, and also information on what we can potentially learn from this. Besides the size argument, the more prominent differences are introduced as we solve thickness-integrated equations in AvaFrame::com1DFA.

Hence the numerical particles represent columns with thickness-averaged quantities. In the computations, they are not assigned a 'size' in terms of area, they have a mass and in an

intermediate step, the particles' mass is interpolated onto the four nearest grid points and from this mass and the cell area a flow thickness is computed. This further implies that the numerical particles cannot overtake each other in the slope-normal direction. However, the particles can move and redistribute according to the acting forces and particles can overtake each other in the slope-parallel direction. As we use a Lagrangian method, tracking of individual particles is straight-forward and if we increase the number of particles, we increase the spatial resolution (amongst just the number of particles, other numerical parameters such as time step and the kernel radius have to be adjusted accordingly). By doing so, we expect to get a more distinct result tracking those particles in the vicinity of the initial AvaNode location. Additionally, as the simulated avalanche is rather 'small', an increased spatial resolution compared to the default setup is desirable.

C14. line 189. At the first glance, I had in mind that the 'vertical' component of the velocity (=normal to the local slope) of the particles in simulations is by construction zero (depth-averaged). But as the reference frame is the absolute Cartesian one (x,y,z), I'm realising that  $v_{z,i}^{s,i}$  is not nil. I think it would be great to show (x,y,z) frame in Fig. 1 and better state all this.

A14. In AvaFrame::com1DFA, the equations are solved using a Lagrangian approach and a local coordinate system in the tangent plane to the surface (which is a 2D manifold embedded in the 3D Euclidean space). This tangent plane is derived using the normal to the surface and the velocity direction at the respective location, see Tonnel et. Al (2023). For the comparison to the AvaNode velocities, the x, y, z components of the numerical particles' velocity in the Cartesian reference frame are used.

**We will update Line 176 to:**

The AvaNodes record position and velocity in a three-dimensional World Coordinate System, where the axes are defined as follows: x points East, y points North, and z points vertically upward. All further analyses, including the comparison between measurements and simulations, are consistently performed in this World Coordinate System. Deviations are first computed separately along each axis in three dimensions (as defined in Eq. (2)) and are then combined into a single magnitude (as defined in Eq. (3)).

And add a description in Figure 1, that the left panel shows the x-y plane.

**C15. lines 218-220. I think this is a nice outcome of your study. I would suggest to put more emphasis on this result if this was a key objective.**

A15. This was not a key objective, this is an outcome. To highlight more the potential of this outcome we will change paragraph [Line 298-301] to:

Interestingly, the optimization results revealed that different observational datasets, such as the AvaNode velocities and radar front positions, lead to distinct yet relatively narrow bands of well-performing parameter combinations. As an outlook, a promising strategy would be to combine these complementary observational constraints in a weighted manner, depending on the specific modelling objective. Such a targeted combination could enable the identification of parameter sets that simultaneously provide good agreement with both particle and front observations, thereby improving the robustness and general applicability of simulation results.

C16. line 225. the values are different from the particle velocity analysis. Could you comment on this. This is an interesting point. However, we cannot exclude the fact that as you are tracking particles that are different (avanodes or simulated one) and different from the true snow granules (see other previous comments) the particle velocities are not representative of the true snow granules in the end. As such, shall we rely more on values coming from the front (radar and simulations) rather than on the values coming from particle velocities?

A16. The radar measurements alone provide us with ranges of suitable parameter sets. However, by combining radar front position data with AvaNode particle velocity data, we gain a more comprehensive insight into the avalanche dynamics. The flow characteristics differ significantly between the avalanche front and the tail, with the front typically governed by more inertial dynamics and the tail by more frictional and slower flow regimes. The AvaNodes, although not identical to natural snow granules, allow us to access valuable information about the behaviour in the tail region, which would otherwise remain unobserved with front-based measurements alone. Therefore, we believe that a combination of both data types,front position and particle velocities,is essential to better capture the spatially varying dynamics of snow avalanches.

C17. line 237-238. OK you are comparing to the suggestion of Gauer (2014) for maximum velocity scaling. But what about the key difference in the curvature between the simple Gauer's prediction and the trend of your results? Could you comment on this? Could it stem from complex interplay with specific topography, or would it be something else.

A17. The observed curvature is an unphysical result of including computations with unacceptable mean square errors in the figure. To improve clarity, we adjusted the selection of simulations shown to include only those with a positional error smaller than that of AvaNode C10. This allows us to better illustrate where the simulated runouts terminate and what maximum velocities are reached. The comparison with the suggestion by Gauer (2014) serves to assess whether both simulations and measurements fall within an expected and physically plausible range.

C18. line 264. Yes, there are key granular processes behind this observation. I'm still not convinced about the generic conclusions we can get from just three AvanNodes with two different density (that are quite high in the end). And again what is the size of the AvaNodes ? Maybe some minimum information about AvaNodes should be added (reference to key previous papers is not enough).

**A18. Did you mean Line 294?**

Please see A2 for intended changes in the revised manuscript.

**C19. line 295. Would it mean that taking into account processes at the grain level for snow avalanche modeling remain secondary?**

A19. While simplified models can be calibrated to match certain aspects of avalanche behaviour, such as achieving a reasonable fit to specific measurements, they still fall short of capturing the full complexity of the flow. For instance, discrepancies between the dynamics at the avalanche front and tail remain unresolved. This suggests that grain-scale processes can play a crucial role in shaping the overall flow structure and should not be overlooked in future model development and validation efforts.

C20. lines 313-314. One single particle only cannot be representative of the whole avalanche process of course, in particular at the end when more and more particle are located at the tail. Be cautious. Again, wouldn't it be better to rely on front position?

A20. Relying solely on the front neglects potentially valuable information about internal processes, such as velocity variations, particle interactions, or flow regime transitions. Tracking individual particles, albeit with limitations on their representativeness, can still reveal these internal dynamics and help to identify trends not visible from the front alone. Ultimately, a more comprehensive understanding of avalanche behaviour may require integrating additional parameters, such as characteristics of the deposition zone, or even snow temperature, to better capture the full complexity of avalanche events.

C21. lines 334-335. Yes, of course. We expect some frictional hysteresis between the front (inertial granular regime) and the tail (much less inertial regime), as well as complicated phase transitions in terms of densities, as known from measurements by Sovilla et al at Vallée de la Sionne.

A21. Thank you very much for this important comment. Indeed, measurements from largescale avalanches, such as those conducted by Sovilla et al. at Vallée de la Sionne, show clear evidence of frictional hysteresis between the avalanche front and tail. It is very interesting to observe similar behaviours even in our small-scale avalanche experiments.

We will overwork the paragraph in Line [301-305] to:

Our optimizations for the avalanche front, based on radar data, and for the tail, based on AvaNode particle data, reveal that the particle and front behaviour cannot be simultaneously reproduced by the same set of flow model parameters and friction relations. This suggests that different flow regimes dominate different parts of the avalanche: the front is governed by more inertial, dynamic processes, while the tail transitions into a slower, more frictiondominated regime. This observation is consistent with the concept of frictional hysteresis and phase transitions in density, as previously reported in large-scale avalanche measurements, such as those by Sovilla et al. at Vallée de la Sionne. Our results highlight that even in smaller avalanches, these complex dynamics are present, emphasizing the need for models that can account for spatially and temporally varying flow behaviours. To further support this finding, we provide a supplementary video that visualizes both best-fit simulations, one optimized for the front and one for the tail, offering an intuitive overview of the differences in particle dynamics and flow structure across the avalanche (Neuhauser et al., 2024).

**C22. section 4.2. I'm not so convinced by the importance of initial and boundary conditions for isolated particles...**

A22. Thank you for this comment. While initial and boundary conditions have been investigated for the "whole" avalanche, the knowledge on isolated particles has not gained much attention. In this work we utilize the "new" particle tracking functionalities to gain a deeper understanding on this topic. Surprisingly the tracking and projection of simulation results shows that this topic may require a more detailed look – although data is sparse, and no general trends are deduced (also because avalanche and particle properties may vary) we see that initial position may have a significant role in resulting velocity or travel distance (at least in the simulations). To highlight these points and the difference between isolated particles and the whole avalanche, we adapted the text accordingly:

The influence of boundary conditions, such as release thickness, release area or topography on simulation results is well known, considering the main avalanche features (Bühler et al., 2011; Bühler et al., 2018). In this paper with particular focus on particle tracking we investigate the interplay between topography and initial position within the release area for isolated particles. To do so, the com1DFA module of AvaFrame has been extended with the presented particle tracking. The implementation of these functionalities allows us for example to project simulation results along the particle trajectories forwards and backwards in time. With this information, one can display and analyse how flow quantities, such as maximum velocity or travel length, develop along potential particle trajectories. In the future, the methodology could allow predictions where and how something will be transported if the starting point is known. As a first application we use this methodology to test whether the interaction of local topography and initial position in the release area determines the resulting maximum velocities. It is important to note that observational data remains sparse and that it is not possible to investigate the relative influence of the different effects, such as comparing the role of initial position of an AvaNode to its density.

**C23. lines 353-356. Not sure to get your statement. As snow granules interact through complex interactions during the avalanche I don't really see how/why their trajectories would be primarily controlled by the initial position in the release area.**

A.23 Thank you for this important remark. We agree that for real snow granules, complex interactions dominate the dynamics throughout the avalanche. Our statement referred specifically to the simulated particles in the model, where particle trajectories are primarily controlled by their initial position in the release area, due to the absence of explicit granular interactions in the simulation framework. We will clarify this point in the text to avoid misunderstandings.

C24. lines 362-362. This kind of clear statement about the differences between simulated particles and Avanodes, and real snow granules should be said much earlier I think.

A24. We refer to A4, where we tried to explain the fundamental differences in the Introduction.

**Editing issues / typos / suggestions**

- title: after having read the whole paper, maybe something like this could be another relevant (if not better?) option for the title: "Particle and front tracking of one single avalanche event from inflow sensors and radar measurements backed-up with simulations".

By adding the new Objectives section, we hope it is now clear that one of the key goals of this work is the development of particle tracking both in experimental settings and within computational avalanche dynamics. This dual focus forms a central aspect of our contribution. Therefore, we would appreciate the opportunity to retain the current title, as we believe it accurately reflects the scope and core aim of the study.

abstract needs a thorough revision. I think the abstract in the present state is too long. It needs to be shortened, more synthetic. I think it has something to do with major concern that we don't really know what is the key objective of the paper.

We will change this accordingly, here is the new abstract:

Understanding particle motion in snow avalanches is crucial for improving the representation of flow dynamics in numerical models. In this study, we develop and apply a general framework for testing and calibrating thickness-integrated (Tonnel et al., 2023) flow models using in-flow sensor data from AvaNodes, radar measurements, and simulations with the com1DFA module of the open-source AvaFrame framework. This includes an implementation of particle tracking functionalities and focuses on assessing a modified Voellmy friction relation.

Radar measurements of the avalanche front and three-dimensional AvaNode trajectories provide a comprehensive observational basis for model comparison. By minimizing the differences between measured and simulated velocities and front positions, we identify parameter sets that achieve high agreement with observed dynamics, yielding deviations below 5–10% in maximum velocity and travel distance. However, the results reveal a trade-off between accurately reproducing particle versus front behaviour, reflecting model limitations and the presence of equifinality in the parameter space.

We also find that the simulated particle velocities are primarily controlled by initial position, contrasting with experimental observations that show more complex particle interactions. These findings underline the need for enhanced model formulations to better capture flow regime transitions and particle-scale effects. Our results highlight the potential of combining multiple measurement types for calibration and future improvements in avalanche modelling.

- line 3 (abstract). I was initially surprised by "thickness integrated". The term "depthaveraged" may refer to a more common semantics... But I've checked Tonnel et al (2023) and saw that there were some explanation about this. I think it would be pertinent to refer to Tonnel et al for this specific choice of the semantics.

Will be changed accordingly.

- line 4 (abstract). I would invert: "the open avalanche framework, named Avaframe."

This suggestion has been considered in the revised version of the abstract.

line 25 (introduction). The start here is quite surprising (if not weird) with this sentence alone. Could you elaborate a bit more and cite some relevant literature?

We will change the introduction towards the description of particles in avalanches. See C1 and A1.

- figure 1, top and bottom right plots: there are issues in the inserted legend (top plot) and x,y labels (bottom plot)... Please revise.

We had a problem with the included Figure as pdf, we changed it to png to avoid problems.

- line 104. Please be consistent along the whole manuscript: use a '-' between 'thickness' and 'integrated' or don't use it but stick to one single option.

Will be changed accordingly.

- caption of figure 6, second line: there is a typo: an empty space is missing after the comma.

Will be changed accordingly.

- caption of Table 3, second line: there is a typo "... delta Z) (Fig. 8) ..." Please fix it.

Will be changed accordingly.

References:

Rastello, M., Marié, J.-L., and Lance, M.: Clean versus contaminated bubbles in a solid-body rotating flow, Journal of Fluid Mechanics, 831, 592–617, 2017.

Bartelt, P. and McArdell, B.: Granulometric investigations of snow avalanches, Journal of Glaciology, 55, 829–833, https://doi.org/10.3189/002214309790152384, 2009

Sampl, P., Zwinger, T.: Avalanche simulation with SAMOS , Annals of Glaciology 38(1), International Glaciological Society, 393–398, 2004

Fischer, J.-T.: A novel approach to evaluate and compare computational snow avalanche simulation , Natural Hazards and Earth System Science 13(6), 1655–1667, 2013

Marks, B. and Einav, I.: A mixture of crushing and segregation: The complexity of grainsize in natural granular flows, Geophysical Research Letters, 42, 274–281, https://doi.org/https://doi.org/10.1002/2014GL062470, 2015.

Marks, B. and Einav, I.: A heterarchical multiscale model for granular materials with evolving grainsize distribution, Granular Matter, 19, 61, https://doi.org/10.1007/s10035-017-0741-6, 2017.

Edwards, Andrew, Rocha, FM, Kokelaar, BP, Johnson, Christopher, Gray, JMNT: Particle-size segregation in self-channelized granular flows, Journal of Fluid Mechanics, Cambridge University Press, 2022

Steinkogler, Walter, Gaume, Johan, Löwe, Henning, Sovilla, Betty, Lehning, Michael: Granulation of snow: From tumbler experiments to discrete element simulations, Journal of Geophysical Research: Earth Surface 120(6), Wiley Online Library, 1107–1126, 2015

Sovilla, B, McElwaine, JN, Köhler, A: The Intermittency Regions of Powder Snow Avalanches , Journal of Geophysical Research (Earth Surface) 123, 2525–2545, 2018

Neuhauser, M., Köhler, A., Wirbel, A., Oesterle, F., Fellin, W., Gerstmayr, J., Dressler, F., and Fischer, J.-T.: Particle and front tracking in experimental and computational avalanche dynamics, https://doi.org/10.5446/68602, 2024.

---

## Author Comment (AC2)

We are grateful for the suggestions of the reviewer and his detailed annotations in the PDF. We will revise the manuscript carefully including the annotations given in the PDF. In the following we give detailed answers (in blue) to the comments.

This is a unique and well organized study of avalanche dynamics, which uses in-flow particle sensors, FMCW radar, and avalanche dynamics simulation. The approach is novel and provides new insight into parameterization of avalanche flow models. This is a fundamental contribution to this field, and should be published after minor revisions. I have some main points to consider below, and have attached an annotated PDF with detailed suggestions.

Thank you very much for your positive and constructive feedback on our manuscript. We are pleased that you consider this work a fundamental contribution to the field of avalanche dynamics and appreciate the novelty of combining in-flow particle sensors (AvaNodes), FMCW radar, and simulation for improved model parameterization.

Following your suggestions, we have carefully addressed all comments and will revise the manuscript accordingly. In particular, we will make the following key changes:

- Deliver an improved description of sensor accuracy, including the radar setup, resolution, and positioning in the methods section.
- Add more detail about the avalanche event itself, including a clearer definition of the avalanche characteristics.
- Clarify issues regarding fracture depth variation, uncertainty estimates, and the combination of measurement results to provide a more comprehensive understanding of the data interpretation.

We hope that these revisions will addres all concerns and improve the clarity and scientific value of the manuscript. We look forward to the continued review process.

Sincerely,
Michael Neuhauser

1) Sensor details. Some additional details are needed about the sensors - where is the radar located? What frequency range and range resolution? How accurate do you expect the avalanche front observations to be? This is important to help the reader interpret the results.

A1) We will revise section 2.1 with more information about the measurement systems and rewrite Line [90-95] to:

The FMCW radar system (mGEODAR) was positioned at Seegrube at 1900 m.a.s.l., approximately 700 m away from the release area on the north-facing slope of the Seilbahnrinne, providing a clear line of sight to the main avalanche path up to the avalanche dam, beyond which the radar view becomes obstructed. The radar operates with a range resolution of 0.375 m per bin and a sampling frequency of 50 Hz, enabling the precise tracking of the avalanche front over time. Based on prior evaluations and controlled experiments, the expected positional uncertainty of the radar-tracked avalanche front is estimated to be approximately ±1–2 m, which corresponds to around five radar range bins. This provides reliable observations for tracking the avalanche front's evolution, especially in the middle section of the path where radar line of sight aligns well with the flow direction. In comparison, the AvaNodes (C07, C09, and C10) record GNSS-based three-dimensional positions at 10 Hz, with a manufacturer-specified horizontal position accuracy of ±2.5 m and a

Doppler-based velocity accuracy of ±0.05 m/s along each axis. While the radar offers high temporal and spatial resolution of the avalanche front, the AvaNodes provide complementary data on particle-level dynamics, particularly in the tail of the avalanche. The combination of both systems offers a more comprehensive view of avalanche dynamics, allowing the study of differing flow regimes along the avalanche body.

2) How big are the AvaNodes? This isn't currently included. How does the size and density compare to estimates of the snow particles in the avalanche? Why were the two different densities chosen? How were the density/size of the actual particles estimated?

A2) We will add a more detailed explanation on the size of the AvaNodes and why we came up with these densities. The used densities are inherent to the prototype design, with efforts focused on minimizing weight while accommodating necessary hardware, power source and casing constraints. A density of 415 kg/m³ represents the lowest achievable value within the current design limitations. With the gained experience and further advancements in using these sensors in avalanche experiments, it is now possible to produce a wider range of densities and likely reduce the minimum density of the AvaNodes.

We will add a paragraph to make this clear, at Line [84]:

While snow particles in avalanches tend to have a density between 100 and 400 kg/m³ during the movement and 250 to 400 kg/m³ in the deposition (Dent et al., 1998), it was the goal to achieve similar densities for the AvaNodes. However, due to design constraints, the minimum achievable density was 415kg/m³.

The AvaNodes are cubelike bodies with an outer length of 16 cm, which is comparable to the typical size of snow granules found in the deposition zone (Bartelt and McArdell, 2009). The inclusion of a higher-density node (688 kg/m³) was intended to introduce variation, facilitating the investigation of potential differences in the behaviour of varying densities during flow and transport.

3) Some discussion of the avalanche type (dry vs wet) and the flow regime that the sensors and simualtions are representing is needed at the beginning of the paper, as the methods are introduced.

Based on meteorological records and observations from the deployment team, the avalanche can be classified as a dry, dense flow avalanche.

We will expand paragraph Line 75 -80 with more information about the avalanche event:

The data sets used in this article originate from an avalanche experiment (number #20220025) that was performed on the 22 of February 2022, at the test site Nordkette, Seilbahnrinne, in Austria. The avalanche was released during avalanche control work after a new snow precipitation event of around 40 cm new snow at Seegrube. Some parts of the avalanche reached the catching dam at 1800 m asl resulting in a maximum altitude difference $\Delta Z$ of 400 m and a projected travel length $\Delta s\_xy$ of 690 m along the main flow direction. More details to this avalanche event is found in (Neuhauser et. al, 2023).

According to the international avalanche classification (avalanche atlas (UNESCO, 1981)), the observed avalanche classifies as A2B1C1D2E2F4G1H1J4, corresponding to: slab avalanches (A2), with a sliding surface within the snow cover (B1) and dry snow (C1) in the

zone of origin, channelled avalanches (D2), dominated by the dense, flowing part (E2) in the zone of transition and mostly fine (F4), dry (G1), clean (H1) deposits in the zone of deposition with intentional human release (J4) within avalanche control work.

4) How was the accumulation in the starting zone estimated?  How about the fracture depth?  It is stated that an interval board was used and estimates included an assessment of wind redistribution.  How was the wind redistribution component estimated?  Was the release volume varied in the simulations?  Seems like this would be a sensitive parameter, similar to the friction and other parameters investigated.  Some discussion is warranted here.

A4. Thank you for your interest. The release area and release thickness were estimated as fixed input parameters for the simulations. These values were derived by combining manual field observations, including measurements from an interval board located near the release area, with meteorological data from a nearby automatic weather station. Although we acknowledge the potential sensitivity of the release volume, in this study we focused on varying the friction parameters while keeping the release volume constant. We agree that a more detailed sensitivity analysis of the release volume would be a valuable addition for future work.

5) An overall summary of the uncertainties in all the estimates used for the assessment would be helpful - for example, what is the uncertainty in the vertical velocity? The avalanche front position?

A5. Thank you for the suggestion. We agree that a summary of uncertainties is important for the interpretation of our results. The uncertainties of the measurement systems used in this study have been addressed in our first paper (Neuhauser et al., 2023). The AvaNode GNSS modules typically exhibit an uncertainty in vertical velocity of approximately ±0.05 m/s, depending on satellite availability. For radar-based front tracking, the positional uncertainty is estimated at around 1–2 meters, which corresponds to approximately 4–5 radar range bins. We will include a summary of these uncertainties in the revised manuscript for clarity.

Please see also A1 where we show the planned update for Section 3.1, so that more information about the used measurements techniques is available.

6) Why was only C10 focused on in the analysis in terms of the error assessment?  Was it possible to find a set of parameters for all 3 AvaNodes that gave a low error?  I realize this might not be possible when including the avalanche front position, but what about just for all 3 AvaNodes?

A6) We tried to point that out in the Discussion section. Line 295-300 and with Figure 4,5 and 6.

In general, it is possible to determine a parameter set for the three AvaNodes as well as for the avalanche front. The challenge is to combine the different measurements. One approach is to take the mean values of all measurements. Alternatively, a weighting scheme could be applied. For example, assigning less weight to C07 due to its density, which does not represent typical snow granules in this avalanche, and assign a higher weight to the avalanche front, as it provides critical impact pressures.

We aimed to highlight that a wide range of parameter sets can reproduce the movement of both the avalanche front and the AvaNodes with low errors, as shown in Figures 4 to 6.

However, the methodology for optimally combining these parameters is such a complex topic that should be the focus of a separate study.

We will change the paragraph Lines [298-301] to:

Interestingly, the optimization results revealed that different observational datasets, such as the AvaNode velocities and radar front positions, lead to distinct yet relatively narrow bands of well-performing parameter combinations. As an outlook, a promising strategy would be to combine these complementary observational constraints in a weighted manner, depending on the specific modelling objective. Such a targeted combination could enable the identification of parameter sets that simultaneously provide good agreement with both particle and front observations, thereby improving the robustness and general applicability of simulation results.

Overall this is an excellent paper!

Thank you for this very encouraging statement.

---

## Author Response (AR2)

Dear Editor,

We thank you very much for your positive feedback and for approving our manuscript for publication. We carefully addressed all the minor issues that were pointed out:

- 1. We ensured consistent notation of all units throughout the manuscript.
- 2. We added the units (degrees) to the slope angle in the color map legend of Figure 2.
- 3. We indicated the correct units for  $\xi$  and  $\tau 0$  in the headers of Figures 3 and 7.

We are grateful for your support and the opportunity to contribute to the NHESS special issue.

Sincerely, Michael Neuhauser